



# Potential Artifacts of Sequential State Estimation Invariants

Carl Wunsch[1]

[1]Department of Earth and Planetary Science, Harvard University, Cambridge MA 02138

**Correspondence:** Carl Wunsch (carl.wunsch@gmail.com)

**Abstract.** In sequential estimation methods often used in general climate or oceanic calculations of the state and of forecasts, observations act mathematically and statistically as forcings as is obvious in the innovation form of the equations. For purposes of calculating changes in important functions of state variables such as total mass and energy, or in volumetric current transports, results are sensitive to mis-representation of a large variety of parameters including initial conditions, prior uncertainty covariances, and systematic and random errors in observations. Errors are both stochastic and systematic, with the latter, as usual, being the most intractable. Here some of the consequences of such errors are first analyzed in the context of a simplified mass-spring oscillator system exhibiting many of the issues of far more complicated realistic problems. The same methods are then applied to a more geophysical barotropic Rossby wave plus western boundary current system. The overall message is that convincing trend and other time-dependent determinations in "reanalyis" like estimates requires a full understanding of both models and observations.

## 1 Introduction. Formalism and Simplified System

Intense scientific and practical interest exists in understanding the time-dependent behavior in the past and future of elements of the climate system. Best estimates of past, present, and future invoke knowledge of both observations and models. These both can involve physical-dynamical, chemical, and biological elements.

Fundamental to understanding many physical systems is analysis of long-term changes in quantities such as energy, enstrophy, total mass, mean concentrations, that are subject to various conservation rules. These elements, absent external perturbations or internal sources or sinks, can be usefully regarded as potential "invariants" of the system[1]. In conventional science, violation e.g., of mass or energy conservation not attributable to specific disturbances, would preclude any claim to understanding of the physics, chemistry, etc. governing the temporal evolution. Observational scientific fields in which time series data are of basic importance thus struggle with inferences from changing observation systems—either or both of changing technology or of spatial and temporal distributions. In climate science particularly, both of these factors determine the ability to determine trends over months, decades and longer.

---

[1]The modifier "potential" is normally omitted here, being implicit as true invariants require the absence of generalized dissipation and external forces. "Invariant" is also used as a label for specific sub-elements such as a fixed current transport.





Much interest also exists in the possibility of trends in major sub-elements of the system—oceanographically for example, in the transports of mass or heat or other properties in major currents such as the Gulf Stream. "Best estimates" of these values

are also made using combinations of kinematic and dynamical models plus observations.

Methods for combining data with models fall into the general category of control theory, in both mathematical and engineering forms, although full understanding is made difficult by the need to combine major sub-elements of different disciplines, including statistics of several types, computer science, numerical approximations, oceanography, meteorology, climate, dynamical systems theory, and the observational characteristics of very diverse instrument types and distributions. Within the control

theory context, distinct goals include "filtering" (what is the present system state?), "prediction" (what is the best estimate of the future state?), and "interval smoothing" (what was the time history over some finite past interval?) and their corresponding uncertainties.

In oceanography, and climate physics and chemistry more generally, a central tool has become what meteorologists call a "reanalysis,"-a sequential estimation method based ultimately on long experience specifically with weather prediction. Partic-

ular attention is called, however, to the paper of Bengtsson et al. (2004) who showed the impacts of observational system shifts on outcomes with some sequential methods. A number of subsequent papers (e.g., Bromwich and Fogt, 2004; Bengtsson et al., 2007; Carton and Giese, 2008; Thorne and Vose, 2010) have noticed difficulties in using "reanalyses" for long-term climate properties sometimes ending with advice—such as "minimize the errors" (and see Wunsch, 2020 for one global application).

For some purposes e.g., short-term weather or other prediction, system failure to conserve mass or energy or enstrophy may

be of no concern—as the time-scale of emergence for measurable consequence of that failure can greatly exceed the forecast time. In contrast, for reconstruction of past states, those consequences can destroy any hope of physical interpretation. In long-duration forecasts with rigorous models, but by definition, no observational data at all, invariants are likely to be preserved, albeit tests of model elements and in particular of accumulating errors, are not possible.

Somewhat notorious, contradictory, results in the public domain (e.g., Hu et al., 2020; Boers, 2021) domain suggest that,

given the technical complexities of a full reanalysis computation, that some simple examples of the known difficulties with sequential reanalysis methods could be helpful. Expert practitioners of the methodology, particularly on the atmospheric side (e.g., Dee, 2005; Cohn 2010; Janjic et al. 2014; and Gelaro et al. 2017) clearly understand the pitfalls of the methodologies, but many of these discussions are couched in the mathematical language of continuous space-time (requiring the full apparatus of functional analysis) and/or the specialized language of atmospheric sciences.[2] The intention here is to produce simple

examples, in discrete time—as used on computers with real data, and with a mainly generic vocabulary. Dee (2005) is close in spirit to what is attempted here.

## 1.1 Some Concepts and Notation

*Models*

Some basic notation is necessary to analyze even the simplest, linear, time-evolving system with data. A fuller account is

given in Wunsch, 2006, hereafter W06, and in many other textbook references. (Notation here is similar to that in W06.). Let

---

[2]Note that the continuous time operator is known as the Kalman-Bucy filter and is not used here.





$\mathbf{x}(t)$ be a state vector in discrete time $t = 0, \Delta t, ...., M_t \Delta t = t_f$. A "state vector" is one that completely describes a physical system evolving according to a *perfect* (here linear) model rule,

$$\mathbf{x}(t + \Delta t, -) = \mathbf{A}(t)\mathbf{x}(t, -) + \mathbf{B}(t)\mathbf{q}(t),\tag{1}$$

where $\mathbf{A}(t)$ is the "state transition matrix". $\mathbf{B}(t)\mathbf{q}(t)$ is a very general representation of boundary conditions and any internal
sources or sinks in which $\mathbf{B}(t)$ simply distributes the time-evolving field, $\mathbf{q}(t)$, amongst state vector elements. $\Delta t > 0$ is a fixed time-step. A minus sign has been entered into the argument—from a control theory convention—to indicate that no data are being used.

Such perfect models do not exist in practice and the system is usefully rewritten as,

$$\tilde{\mathbf{x}}(t + \Delta t, -) = \mathbf{A}(t)\tilde{\mathbf{x}}(t, -) + \mathbf{B}(t)\mathbf{q}(t) + \mathbf{\Gamma}(t)\tilde{\mathbf{u}}(t).\tag{2}$$

A tilde, ~, indicates that the solutions to Eq. (2) are at best an approximation to or estimate of the true state vector. $\mathbf{\Gamma}(t)\mathbf{u}(t)$, a flexible structure is introduced as the *unknown* elements and corrections to $\mathbf{B}(t)\mathbf{q}(t)$ for boundary/initial conditions, internal parameterizations, and forcing generally. Eq. (2) will be referred to below as the "prediction model," as it is used in practice to make the best prediction at any future time—given the immediate past best-estimate.[3] In such a calculation, $\tilde{\mathbf{u}}(t) = 0$, as it is otherwise unknown. In many circumstances (e.g., Brown and Hwang, 1997, W06), Eqs. (1 or 2) are linearized about some
reference state. That $\mathbf{A}(t)$ is itself then, and always, subject to significant error is a very important point, but that possibility renders the problem non-linear, and for present purposes the implications and approaches are set aside.[4]

Time-evolving systems require initial conditions, $\tilde{\mathbf{x}}(0)$, having some known or assumed error (uncertainty), written for linear systems as a covariance matrix,

$$\mathbf{P}(0) = \left\langle [\tilde{\mathbf{x}}(0) - \mathbf{x}(0)][\tilde{\mathbf{x}}(0) - \mathbf{x}(0)]^T \right\rangle\tag{3}$$

and with the further, sometimes wholly implicit, assumption that the mean error $\langle \tilde{\mathbf{x}}(0) - \mathbf{x}(0) \rangle = \mathbf{0}$. The brackets denote an expected value, whether theoretical or estimated. Uncertainties generally determine the utility of any solution of any problem. Part of the estimation problem is to cope with the likelihood that $\mathbf{P}(0)$ itself is not wholly accurate, and with implications depending on how long the system "remembers" its initial conditions (typically a function of $\mathbf{A}$).

Let $\mathbf{P}(t, -)$ represent the error covariance (uncertainty) of the prediction, $\tilde{\mathbf{x}}(t, -)$ :

$$\mathbf{P}(t, -) = \left\langle (\tilde{\mathbf{x}}(t, -) - \mathbf{x}(t))(\tilde{\mathbf{x}}(t, -) - \mathbf{x}(t))^T \right\rangle\tag{A1moved}$$
$$= \mathbf{A}(t)\mathbf{P}(t - \Delta t)\mathbf{A}(t)^T + \mathbf{\Gamma}(t - \Delta t)\mathbf{Q}(t - \Delta t)\mathbf{\Gamma}(t - \Delta t)^T,$$

a matrix Riccati equation, which is just the sum of the error covariance propagated from the predicted estimate, plus that generated by unknown forcing, etc., elements, $\mathbf{u}(t)$, with $\mathbf{Q}(t) = \left\langle \mathbf{u}(t)\mathbf{u}(t)^T \right\rangle$, the bracket defining an ensemble average.

---

[3] A tilde could be placed over all the elements, including $\mathbf{A}, \mathbf{B}, ...$ but here, that labelling, for notational simplicity, is restricted to the state variables.

[4] The existence and use of the information contained in such *a priori* models, kinematic, thermodynamic, biological, chemical, and otherwise distinguishes this approach from some attempts to use machine learning to deduce a fully *a posteriori* model. The result of Pitandosi (2018) is thus a challenging one.





(See any of numerous textbooks cited above.) $\mathbf{Q}(t)$ represents the error covariance of $\mathbf{q}(t)$, and thus of $\mathbf{u}(t)$. Inaccuracies

in this equation are discussed by Konstantinov et al. (1993), Zhou et al. (2009) and $\mathbf{P}$ must always itself be regarded as an estimate, not "truth." Note that only precision, not accuracy, is the subject here.

A useful conceptual generalization of these methods is to create ensembles of solutions e.g., generated by random selection of different initial conditions from the probability density of the initial conditions (e.g., Evensen, 2009) and then using the results to calculate variances of the corresponding solutions. Difficulties lie with the very large number of elements subject

to random and systematic errors, choice of the correct probability densities, and the usual very small number of ensemble members feasible to compute relative to the dimension e.g., of $\mathbf{x}(t)$. For trend determination accurate knowledge of the overall uncertainties remains important.

*Data*

Suppose now that at time $t = \tau > 0$ some data are available, written generally, but linearly, as,

$$\mathbf{y}(\tau) = \mathbf{E}(\tau)\mathbf{x}(\tau) + \mathbf{n}(\tau), \tag{4}$$

where $\mathbf{n}(\tau)$ is usually *assumed* to be a zero-mean unimodally distributed noise process in the observations, with known covariance matrix, $\mathbf{R}(\tau)$, and which is often time-dependent. (Non-linear observations, for example that of a speed, require special treatment.) Observation matrix $\mathbf{E}(\tau)$, possibly itself having inaccuracies, appropriately distributes the elements of $\mathbf{x}(\tau)$ making up the observations, and which can range from observation of a single element, $x_j(\tau)$, to some arbitrarily complicated

linear combination of different elements (e.g., weighted averages or differences).

Formally, one can deduce another estimate of $\mathbf{x}(\tau)$ directly from Eq. (4) as,

$$\tilde{\mathbf{x}}(\tau, y) = \mathbf{E}(\tau)^{+}\mathbf{y}(\tau) \pm \mathbf{n}_E(\tau), \tag{5a}$$

$$\mathbf{P}_E(\tau) = \left\langle [\tilde{\mathbf{x}}(\tau, y) - \mathbf{x}(\tau)][\tilde{\mathbf{x}}(\tau, y) - \mathbf{x}(\tau)]^T \right\rangle, \tag{5b}$$

where $\mathbf{E}(\tau)^{+}$ is a generalized inverse deduced from standard, *static,* linear inverse methods using appropriate row and column

scaling, and would be accompanied by an uncertainty $\mathbf{P}_E(\tau)$ and a resolution analysis (e.g., W06). Such static, fixed-time, calculations for time-dependent systems are uncommon, as $\mathbf{E}(\tau)^{+}$ usually has a vast unknown nullspace in $\mathbf{P}_E(\tau)$, dependent upon how comprehensive and accurate the data are.

*Combining Data and Models*

Suppose, and as is commonplace in both numerical weather prediction and in reanalyses, that the prediction model is used

to forecast the state at time $\tau$, written as $\tilde{\mathbf{x}}(\tau, -)$ from a previous estimated state. Given the initial condition error $\mathbf{P}(0)$, a straightforward calculation produces an expected error of the forecast, $\mathbf{P}(\tau, -)$ from above.

If data also exist at time $\tau$, then a linear inversion, if carried out as in Eq. (5a), provides another estimate of the state, with its own uncertainty, dependent upon $\mathbf{R}(\tau)$ and the structure of $\mathbf{E}(\tau)$. Evidently, a better estimate than either alone is to combine them in a weighted average, inversely proportional to their uncertainties, as is conventional in recursive least-squares (e.g.,





Lawson and Hanson, 1995; W06) rewritten with minor manipulation as,

$$\tilde{\mathbf{x}}(\tau) = \tilde{\mathbf{x}}(\tau, -) + \mathbf{K}(\tau)\left[\mathbf{y}(\tau) - \mathbf{E}(\tau)\tilde{\mathbf{x}}(\tau, -)\right]. \tag{6}$$

The "gain" matrix is,

$$\mathbf{K}(\tau) = \mathbf{P}(\tau, -)\mathbf{E}(\tau)^T \left[\mathbf{E}(\tau)\mathbf{P}(\tau, -)\mathbf{E}(\tau)^T + \mathbf{R}(\tau)\right]^{-1}. \tag{7}$$

In this form, $\mathbf{K}$ is the "Kalman (1960) gain" and the operation is the "Kalman filter" and which includes, for discrete time, the
uncertainty of the combined estimate,

$$\begin{aligned} \mathbf{P}(\tau) &= \mathbf{P}(\tau, -) - \mathbf{K}(\tau)\mathbf{E}(\tau)\mathbf{P}(\tau, -) \\ &= \mathbf{P}(\tau, -) - \mathbf{P}(\tau, -)\mathbf{E}(\tau)^T \left[\mathbf{E}(\tau)\mathbf{P}(\tau, -)\mathbf{E}(\tau)^T + \mathbf{R}(\tau)\right]^{-1}\mathbf{E}(\tau)\mathbf{P}(\tau, -), \end{aligned} \tag{8}$$

a matrix Riccati equation which is again a result of recursive least-squares.[5] Textbooks prove that the norm, $\|\mathbf{P}(\tau)\| \leq \|\mathbf{P}(\tau, -)\|$, that is, if used realistically, the data cannot worsen the error in the forecast, but can potentially improve it, perhaps
greatly, depending upon $\mathbf{E}(\tau), \mathbf{R}(\tau)$. (A tilde can sensibly be placed on $\mathbf{P}, \mathbf{R}$, but is omitted here.)

   For all these and related methods, a potentially very important, but implicit, assumption is $\left\langle \mathbf{n}(t)\mathbf{n}(t')^T \right\rangle = \mathbf{0}, t \neq t'$, that is observational noise is uncorrelated over time. Similarly, $\left\langle \mathbf{u}(t)\mathbf{u}(t')^T \right\rangle = \mathbf{0}, t \neq t'$ and $\left\langle \mathbf{n}(t)\mathbf{u}(t')^T \right\rangle = \mathbf{0}$. If the assumptions fail, a general approach is to model the structures of $\mathbf{n}(t), \mathbf{u}(t)$ as part of the problem—essentially augmenting the state vectors.

*Innovation Forms*

   A slight modification of the system is to combine Eqs. (2) and (6) into the "innovation" forms,

$$\tilde{\mathbf{x}}(t + \Delta t) = \mathbf{A}(t)\tilde{\mathbf{x}}(t, -) + \mathbf{B}(t)\mathbf{q}(t) + \mathbf{K}(t)\left[\mathbf{y}(t) - \mathbf{E}(t)\mathbf{x}(t)\right], \tag{9}$$

again setting the unknown $\mathbf{\Gamma}(t)\mathbf{u}(t) = 0$, or,

$$\tilde{\mathbf{x}}(t + \Delta t) = \mathbf{A}_1(t)\tilde{\mathbf{x}}(t, -) + \mathbf{B}(t)\mathbf{q}(t) + \mathbf{K}(t)\mathbf{y}(t), \tag{10}$$

$\quad \mathbf{A}_1(t) = \mathbf{A}(t) - \mathbf{K}(t)\mathbf{E}(t),$

(Goodwin and Sin, 1984, P. 251), whose importance is that both show explicitly that *data introduction acts as an analogue of externally imposed forcing*.

## 1.2   Simple Example: Mass-Spring Oscillator

For a simple, intuitively accessible analogue system, consider the mass-spring oscillator, following any of McCuskey, 1959,
Goldstein, 1980, W06, Strang, 2007) in the conventional continuous time formulation of simultaneous differential equations.

---

[5]The history of the Kalman filter dates to the 19th Century. See Lauritzen, 1981.




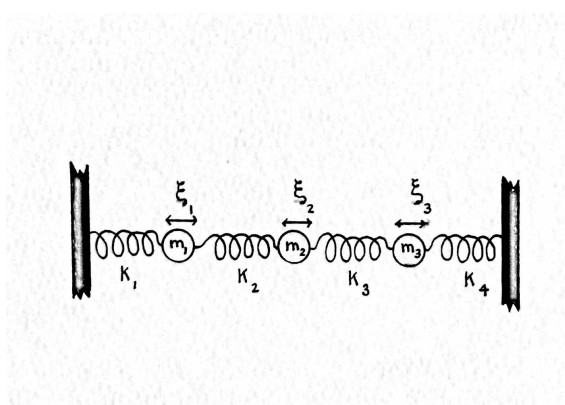

**Figure 1.** Mass-spring oscillator system used as a detailed example. Although the sketch is slightly more general, here all masses have the same value, $m$, and all spring constants and Rayleigh dissipation coefficients $k, r$ are the same.

This choice is obviously somewhat arbitrary, but is both intuitively accessible, and contains sufficient structure to permit the discussion e.g., of spatial average data, fixed mass positions, etc.

Three identical masses, $m = 1$, are connected to each other and to a wall at either end by springs of identical constant, $k$ (Fig. 1). Movement is damped by a Rayleigh friction coefficient, $r$. Generalization to differing- masses, spring constants, and dissipation coefficients is straightforward. Displacements of each mass are $\xi_i(t)$, $i = 1, 2, 3$. The linear Newtonian equations of coupled motion are,

$$m\frac{d^2\xi_1}{dt^2} + k\xi_1 + k(\xi_1 - \xi_2) + r\frac{d\xi_1}{dt} = q_{c1}(t) \tag{11a}$$

$$m\frac{d^2\xi_2}{dt^2} + k\xi_2 + k(\xi_2 - \xi_1) + k(\xi_2 - \xi_3) + r\frac{d\xi_2}{dt} = q_{c2}(t) \tag{11b}$$

$$m\frac{d^2\xi_3}{dt^2} + k\xi_3 + k(\xi_3 - \xi_2) + r\frac{d\xi_3}{dt} = q_{c3}(t). \tag{11c}$$

This second-order system is reduced to a canonical form of coupled first-order equations by introduction of a continuous time state vector, the column vector,

$$\mathbf{x}_c(t) = [\xi_1(t), \xi_2(t), \xi_3(t), d\xi_1/dt, d\xi_2/dt, d\xi_3/dt]^T, \tag{12}$$

where superscript $T$ denotes the transpose. Note the mixture of dimensional units in the elements of $\mathbf{x}_c(t)$, identifiable with the Hamiltonian variables of position and momentum. Then Eqs. (11) become (setting $m = 1$, or dividing through by it),

$$\frac{d\mathbf{x}_c(t)}{dt} = \mathbf{A}_c\mathbf{x}_c(t) + \mathbf{B}_c\mathbf{q}_c(t), \tag{13}$$





where

$$
\mathbf{A}_c = \left\{ \begin{array}{cccccc}
0 & 0 & 0 & 1 & 0 & 0 \\
-0 & 0 & 0 & 0 & 1 & 0 \\
0 & 0 & 0 & 0 & 0 & 1 \\
-2k & k & 0 & -r & 0 & 0 \\
k & -2k & k & 0 & -r & 0 \\
0 & k & -2k & 0 & 0 & -r
\end{array} \right\} = \left\{ \begin{array}{cc}
\mathbf{0}_3 & \mathbf{I}_3 \\
\mathbf{K}_c & \mathbf{R}_c
\end{array} \right\},
\tag{14}
$$

defining the 3x3 block matrices, $\mathbf{K}_c, \mathbf{R}_c$ symmetric and diagonal respectively, and are constant. $\mathbf{B}_c$ distributes inputs, $\mathbf{q}_c = [q_{c1}, ..., q_{c6}]^T$,
variously amongst the six sub-equations. Putting e.g., $r = 0.5, k = 30$, $\mathbf{A}$ is full-rank with 3 pairs of complex conjugate eigen-
values, but non-orthonormal right eigenvectors. These parameter values are generally used throughout. Here, and in what fol-
lows, the system is notationally simplified by using time-constant $\mathbf{A}, \mathbf{B}$. The physics and mathematics of such small oscillations
is discussed in most classical mechanics textbooks and is omitted here. Although a non-dimensionalization is straightforward,
what follows is left in dimensional form to make the results most intuitively accessible.

*Energy*

Consider now an energy principle. Define, without dissipation ($\mathbf{R}_c = \mathbf{0}$), or forcing,

$$
\mathcal{E}_c(t) = \frac{1}{2} \left[ \left( \frac{d\boldsymbol{\xi}}{dt} \right)^T \left( \frac{d\boldsymbol{\xi}}{dt} \right) - \boldsymbol{\xi}^T \mathbf{K}_c \boldsymbol{\xi} \right]
\tag{15}
$$

$$
\frac{d\mathcal{E}_c(t)}{dt} = \frac{1}{2} \frac{d}{dt} \left[ \left( \frac{d\boldsymbol{\xi}}{dt} \right)^T \left( \frac{d\boldsymbol{\xi}}{dt} \right) - \boldsymbol{\xi}^T \mathbf{K}_c \boldsymbol{\xi} \right] = 0
\tag{16}
$$

the sum of the kinetic and potential energies (the minus sign compensates for the negative definitions in $\mathbf{K}_c$) and is here a
Hamiltonian. The non-diagonal elements of $\mathbf{K}_c$ redistribute the potential energy amongst the masses through time.

With finite dissipation and forcing,

$$
\frac{d\mathcal{E}_c(t)}{dt} = \left( \frac{d\boldsymbol{\xi}}{dt} \right)^T \mathbf{R}_c \left( \frac{d\boldsymbol{\xi}}{dt} \right) + \frac{d\boldsymbol{\xi}}{dt}^T \mathbf{B}_c \mathbf{q}(t).
\tag{17}
$$

$d\mathcal{E}_c(t)/dt = 0$, if the forcing and dissipation vanish (see e.g., Cohn 2010, for a formal discussion of such continuous time
systems.)

*Discrete Version*

Write Eq. (1) at constant, discrete, time intervals, $\Delta t$, using an Eulerian time-step in the same form,

$$
\mathbf{x}(t + \Delta t) = \mathbf{A}\mathbf{x}(t) + \mathbf{B}\mathbf{q}(t), t = m\Delta t, m = 0, 1, 2, ....
\tag{18}
$$

$$
\mathbf{A} = \mathbf{I}_6 + dt\mathbf{A}_c,
\tag{19}
$$




and the prediction model is unchanged except now,

$$
\mathbf{A} = \left\{ \begin{array}{cccccc}
1 & 0 & 0 & \Delta t & 0 & 0 \\
-0 & 1 & 0 & 0 & \Delta t & 0 \\
0 & 0 & 1 & 0 & 0 & \Delta t \\
-2k\Delta t & k\Delta t & 0 & (1-r)\Delta t & 0 & 0 \\
k\Delta t & -2k\Delta t & k\Delta t & 0 & (1-r)\Delta t & 0 \\
0 & k\Delta t & -2k\Delta t & 0 & 0 & (1-r)\Delta t
\end{array} \right\}
\tag{20}
$$

$$
= \left\{ \begin{array}{cc}
\mathbf{I}_3 & \Delta t \mathbf{I}_3 \\
\Delta t \mathbf{K}_c & \mathbf{I}_3 + \Delta t \mathbf{R}_c
\end{array} \right\}
\tag{21}
$$

An example for the nearly dissipationless, unforced, example of the oscillator solution, from the discrete formulation is shown in Fig. 2 for elements of $x_i(t)$. Non-zero values here arise only from the initial conditions, $\mathbf{x}(0) = [1, 0, 0, ...]^T$, and which necessarily involve specifying both positions and their rates of change. A small amount of dissipation was included to stabilize the particularly simple numerical scheme. For the choice of the discrete state vector, the energy rate of change, (Fig. 2), is formally identical to that in the continuous case,

$$
\frac{\mathcal{E}_c(t) - \mathcal{E}_c(t - \Delta t)}{\Delta t} = \left( \frac{d\boldsymbol{\xi}}{dt} \right)^T \mathbf{R}_c \left( \frac{d\boldsymbol{\xi}}{dt} \right) + \frac{d\boldsymbol{\xi}}{dt}^T \mathbf{B}_c \mathbf{q}(t).
\tag{22}
$$

$\mathcal{E}_c(t)$ and the potential and kinetic energies through time are also shown. The basic oscillatory nature of the state vector elements is plain, and the decay time is also visible.

The total energy declines by about 2% in an initial transient and then stabilizes with small numerical oscillations at about 5000 time steps. Kinetic energy is oscillatory as energy is exchanged with the potential component. (Unlabelled $y-$axes in plots are non-dimensional variables.)

### 1.2.1 Mass-Spring Oscillator with Observations

If the innovation form of the evolution Eq. (9) is used, the energy change becomes,

$$
\frac{\mathcal{E}_c(t) - \mathcal{E}_c(t - \Delta t)}{\Delta t} \approx
\tag{23}
$$

$$
\left( \frac{d\boldsymbol{\xi}}{dt} \right)^T \mathbf{R}_c \left( \frac{d\boldsymbol{\xi}}{dt} \right) + \frac{d\boldsymbol{\xi}}{dt}^T \mathbf{B}_c \mathbf{q}(t) + \frac{d\mathbf{x}(t)}{dt}^T \mathbf{K}(t) [\mathbf{y}(t) - \mathbf{E}(t)\mathbf{x}(t)],
$$

showing explicitly the influence of the observations on the computed energy. With intermittent observations and/or with changing structures, $\mathbf{E}(t)$, then $\mathcal{E}_c(t)$ will undergo forced abrupt changes—as expected.

Given the very large number of potentially erroneous elements in any choice of model, data, and data distributions, and the ways in which they interact when integrated through time, a comprehensive discussion even of the 6-element state vector mass-spring oscillator system is difficult. Instead, some simple examples exploring primarily the influence of data density on the state estimate and of its mechanical energy are described. One can experiment with the model and its time-constants, model



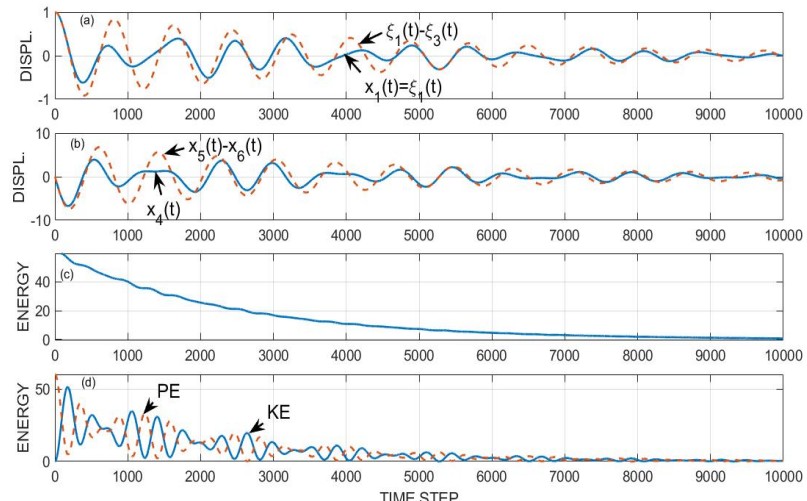

**Figure 2.** The unforced case, initial condition vanishing except for $x_1(t) = 1$. Natural frequency and decay scale are apparent. (a) $x_1(t) = \xi_1(t)$ (solid) and $\xi_1(t) - \xi_2(3)$(dashed). (b) $d\xi_1/dt = \dot{\xi}_4$ and $\dot{}_4 - \dot{}_6$ (dashed). (c) $\mathcal{E}(t)$ showing decay scale from the initial displacement. (d) Kinetic energy (solid) and potential energy making up $\mathcal{E}(t)$.

time-step, accuracies and corresponding covariances of initial conditions, boundary conditions, data etc. The basic problems of any linear system already emerge in this simple example.

Consider, using the same $k, r, \Delta t = .001$ to represent "truth" where the forcing $\mathbf{Bq}(t) = q_1(t) = 0.1\cos(2\pi t/(2.5T_{diss})) +$ 205 $\varepsilon(t)$, that is, only mass 1 is forced in position, and with a low frequency not equal to one of the natural frequencies. $T_{diss} = 1/r$, is the dissipation time. $\varepsilon(t)$ is a white noise element. Initial condition is $\xi_1(0) = 1$, all other elements vanishing; see Fig. 3. Accumulation of the influence of the stochastic element in the forcing clearly depends upon details of the model time-scales and if $\varepsilon(t)$ were not white noise, on its spectrum as well. In all cases, the cumulative effect of a random forcing will have the nature of a random walk—with details dependent upon the forcing structure, as well as the memory elements of the model time 210 scales.

The prediction model (Figs.4, 5) has correct initial conditions and $\mathbf{A}, \mathbf{B}$ matrices, but is forced by the deterministic component with 1/2 the correct amplitude, and with the stochastic component being treated as fully unknown—replaced by its zero mean The added noise in the measurements has a standard deviation of $0.2$ of the total forcing, the latter standard deviation including that of the deterministic contribution.

*Near-Perfect Observations: Two Times and Multiple Times*

To demonstrate the most basic problem of energy, consider a nearly-perfect observation of all 6 positions at two times $\tau_{1,2}$ as displayed in Fig. 4 with $\mathbf{E} = \mathbf{I}_6$ No observational null-space exists. Although the new estimate of the state vector is an improvement over that from the pure forecast, any effort to calculate a trend in energy of the system will fail unless very





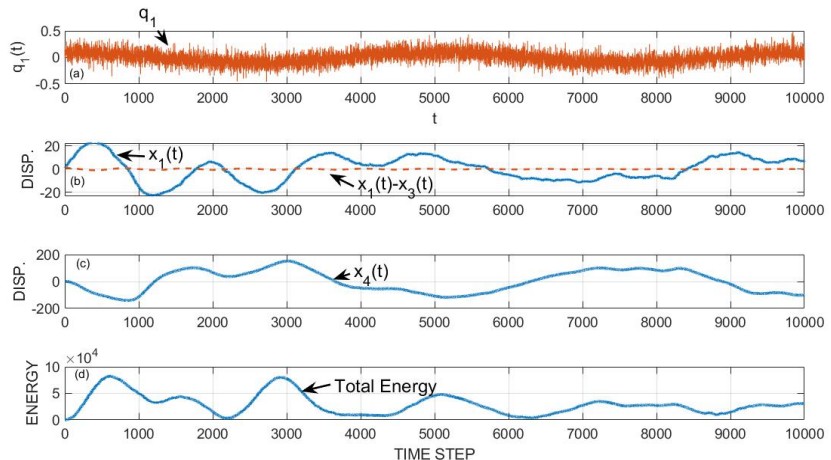

**Figure 3.** Forced version of the same oscillator system as in Fig. 2. Forcing is a low-frequency periodic sinusoid plus a pure white noise disturbance at every time-step in mass 1 position alone. (a) The forcing, $q_1$, white noise plus the visible low frequency sinusoid; (b) $x_1(t) = \xi_1(t), x_1(t) - x_3(t)$; (c) $x_4(t) = dx_1/dt = \dot{\xi}_1(t)$. (d) Total energy through time, $\mathcal{E}(t)$. Energy varies with the random walk arising from $\varepsilon(t)$ as well as from the deterministic forcing. Values are non-dimensional.

careful attention is paid to correcting for the invariant violation at the time of the observation. Fig. 5 shows the results when
observations occur in clusters having different intervals between the measurements. Visually, the displacement and energy have a periodicity imposed by the observation time-intervals and readily confirmed by fourier analysis.

*Quadratic Variability*

In a linear system, a Gaussian assumption for the dependent variables is commonly appropriate. By focussing here on the quadratic invariant of energy, the variables become $\chi^2$ distributed. Thus the $\xi_i^2, \dot{\xi}_i^2$ have such distributions, but with differing
means and variances, and with potentially very strong correlations, so that they cannot be regarded as independent variables. Determining the uncertainties of the six uncertain covarying elements making up $\mathcal{E}(t)$ involves some intricacy. A formal analysis can be made of the resulting probability distribution for the sum in $\mathcal{E}(t)$, involving non-central $\chi^2$ distributions (Imhof, 1961, Sheil and O'Muircheartaigh, 1977, Davies, 1980). In view of the purpose and simplicity of this example however, an estimate of the uncertainty was made by simply generating 50 different versions of the observations, differing in the particular
choice of noise value in each one and tabulating the resulting range. These uncertainties can be used to calculate e.g., the significance of any apparent trend in $\mathcal{E}(t)$ and although the result is not displayed here, use of reliable uncertainties can make an obvious important change in any physical inferences about the state. In these examples, the observational errors are intentionally made relatively small, with no implications for what could be the case in geophysically realistic cases.


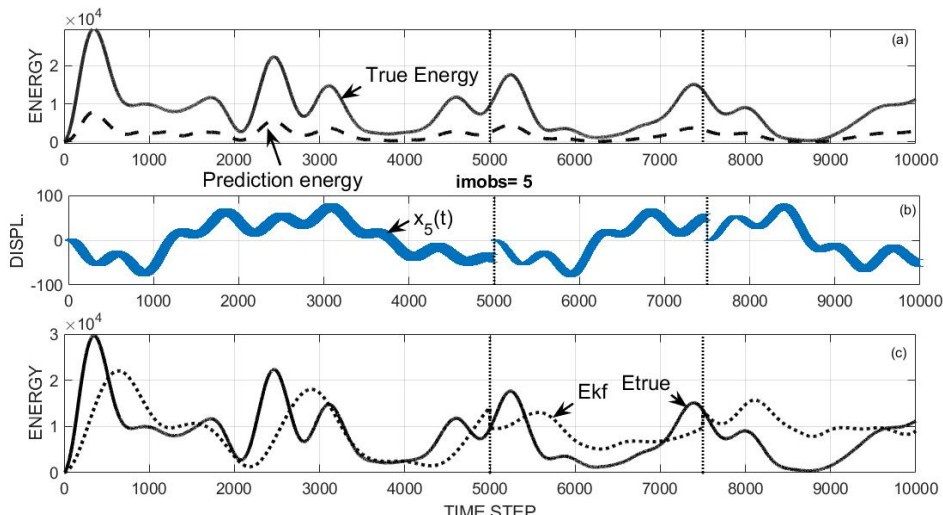

**Figure 4.** (a) Energy for the 3-mass-spring oscillator system ($\mathcal{E}(t)$) and for the prediction model showing the lower energy in the latter. Vertical lines are at the time step when observations become available. (b) Estimated position for velocity in the first mass ($d\xi_1/dt = x_4(t)$) from the Kalman filter and showing the jump at the two times where there are complete *near-perfect* data. Standard error bar is shown from $\mathbf{P}(t)$. (c) $\mathcal{E}(t)$ and $\tilde{\mathcal{E}}(t)$ from the Kalman filter and showing the jumps at the observation times as well as the deviations following the observations.

Notice that even in the observation interval, the estimated mechanical energy remains too low. This bias error is a systematic
one owing to the availability of observations only of the velocity of one of the masses. Even if the observations are made perfect ones (not shown), this bias error in the energy persists (see e.g., Dee, 2005).

As seen in the figure, with full-rank, near-perfect observations the elements of $x_i(t)$ and the total energy are forced to nearly the correct values at the two observation times, $\tau_i$, but do diverge in following times.

*A Fixed Position*

Exploration of the dependencies of energies of the mass-spring system is interesting and a great deal more can be said. Turn however, to a somewhat different invariant: suppose that one of the mass positions is fixed, but with value unknown to the analyst. A significant literature exists devoted to finding changes in scalar quantities such as global mean atmospheric temperatures, or oceanic currents, with the Atlantic Meridional Overturning Circulation (AMOC) being a favorite focus. These quantities are typically sub-elements of complicated models involving very large state vectors. With this very simple mass-
spring oscillator system, it is useful to consider a situation in which an element is a constant, an invariant, but which must be determined from the sequential estimation procedure.

Using the same situation as above, added constraints, that $x_3(t) = \xi_3(t) = 2, x_6(t) = d\xi_3(t)/dt = 0$, that is, an unmoving, fixed displacement in mass 3, are used in computing the true state vector. The observations are the velocity of moving mass

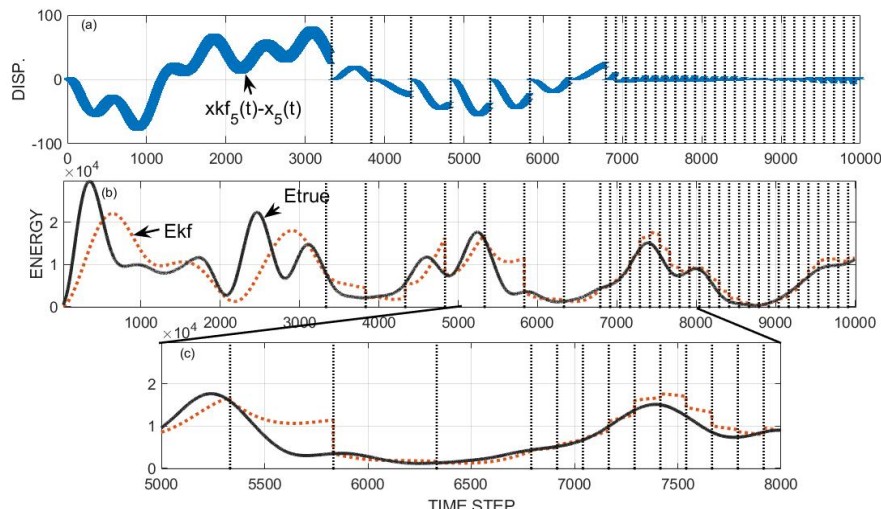

**Figure 5.** (a) $\tilde{x}_5(t) - x_5(t)$ and the same as Fig. 4b except with the observations shown at the times of the vertical dotted lines.(b) Estimated energy in the Kalman filter estimate when observations are available at times of the vertical dotted lines. (c) Expanded portion of (c). Note that the observational errors were here purposely made comparatively small relative to the signals.

$i = 2$, with similar noise in the interval shown in Fig. 6 . The question is whether one can infer accurately that $\xi_3(t)$ is a constant

through time? Note that the fixed displacement means that the potential energy can never vanish. The resulting estimate for the position, $\tilde{\xi}_3(t)$, is shown in Fig. 6 and includes a significant error at all times.

Position variation occurs even during the data dense period and arises both from the entry of the data and the noise in the observations of $d\xi_2(t)/dt$. An average taken over the two-halves of the observation interval might lead to the erroneous conclusion that a decrease had taken place. Such an incorrect inference can be precluded by appropriate use of the computed

uncertainties (Fig. 6).

*Observations of Averages*

Consider now a set of observations of the average of the position of masses 2 and 3, and of the average velocity of masses 1 and 2, mimicking the type of observations that might be available in a realistic setting. Again for simplicity, the observations are very accurate and occur in the two-different sets of periodic time intervals The results are in Fig. 7. Position estimates shown

are good, but not perfect, as is also true for the total energy. Visually, it is clear that the energy estimate carries oscillatory power with the periodicity of the oncoming observations intervals and appears in the spectral estimate (not shown) with excess energy in the oscillatory band and somewhat low energy at the longest periods. Irregular spacing would introduce a potentially complex spectrum in the result.

A more general discussion of nullspaces involves that of the weighted $\mathbf{P}(\tau, -)\mathbf{E}^T$ appearing in the Kalman gain. If $\mathbf{E}$ is

the identity, and $\mathbf{R}(\tau)$ has sufficiently small norm, all elements of $\mathbf{x}(\tau)$ are resolved. If the noise is uniform in all elements




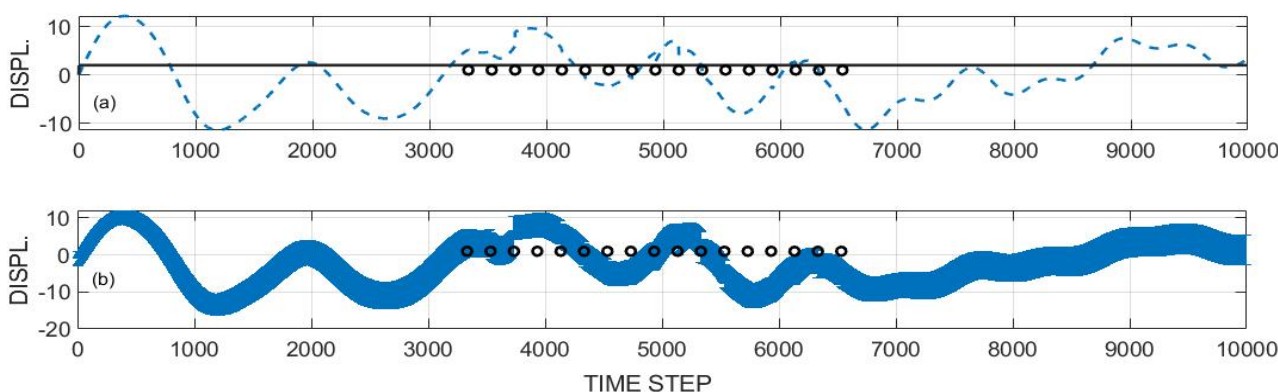

**Figure 6.** (a) Correct value of the constant displacement $\xi_3(t)$ (solid line), and the estimated value from the KF calculation (dashed line). Dots are the observation times. (b) Difference $\tilde{\xi}_3(t) - \xi_3(t)$ and one standard error bar computed from the matrix Riccati equation.

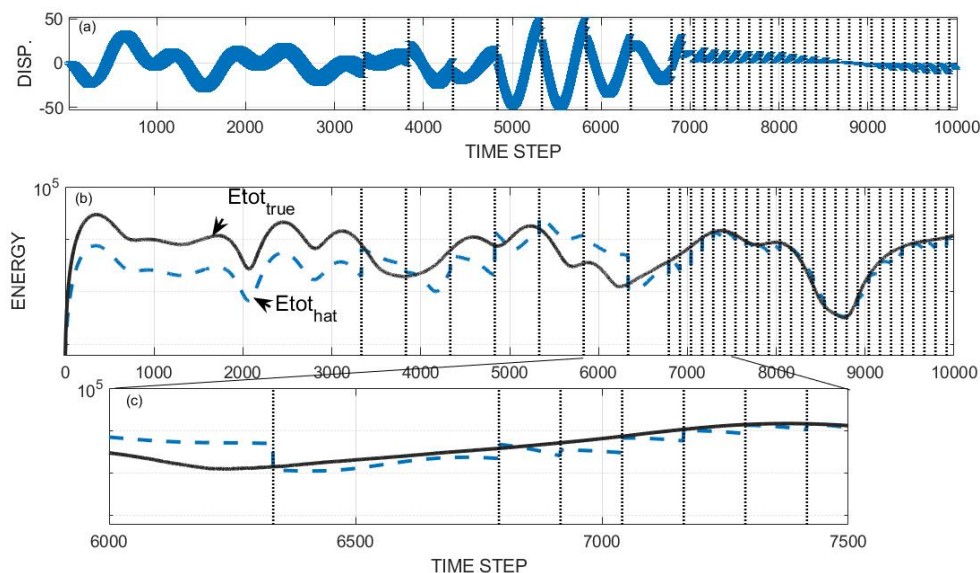

**Figure 7.** (a) Results for position estimate difference $\tilde{x}_5(t) - x_5(t)$ with standard error from the KF when observations were of the average of the two positions $x_2(t), x_3(t)$ and the two velocities, $x_4(t), x_5(t)$ at the times shown. (b) Total energy corresponding to the situation in (a). (c) Expanded portion of (b) showing the artificial periodicity in energy from the combination with observations.




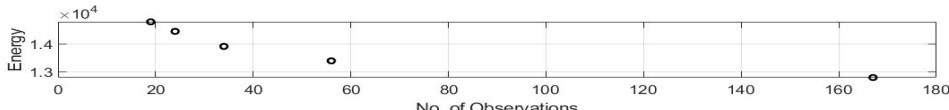

**Figure 8.** RMS difference $\tilde{\mathcal{E}} - \mathcal{E}true$ as a function of the number of data points in the time interval used.

of $\mathbf{y}(\tau)$, the resolution analysis of the observations is also uniform and uninteresting. In the present case, with $\mathbf{E}$ having two rows, corresponding to the observations of the averages of two-mass positions and of two velocity positions, the resolution analysis is more structured. With

$$\mathbf{E} = \left\{ \begin{array}{cccccc} 0 & 1/2 & 1/2 & 0 & 0 & 0 \\ 0 & 0 & 0 & 1/2 & 1/2 & 0 \end{array} \right\} \tag{24}$$

a singular value decomposition $\mathbf{E} = \mathbf{U}\mathbf{S}\mathbf{V}^T = \mathbf{U}_2\mathbf{S}_2\mathbf{V}_2^T$, produces two non-zero singular values, and $\mathbf{U}_2$ etc. carries the first two columns of the matrix. At rank 2, the resolution matrices $\mathbf{T}_U, \mathbf{T}_V$ based on the $\mathbf{U}, \mathbf{V}$ vectors respectively and the standard solution covariances are easily computed (W06). $\mathbf{A}$ distributes information about the partially determined $x_i$ throughout all masses via the dynamical connections as contained in $\mathbf{P}(\tau)$. Bias errors require specific, separate analysis.

The innovation form of equations provides a convenient analysis method for determining the memory duration of varying
observations. Here a Green function discussion has been placed in Appendix C.

### 1.2.2   Varying Data Density

As was conspicuous above, data density in time influences the accuracy of estimates of $\mathcal{E}(t)$. Consider the behavior of the energy estimate as the density of observations varies in time. Fig. 8 displays the RMS difference between the estimated energy over the observation intervals (including non-observation times) as a function of the number of data points included. Compare
Fig. 5.

Similar results will apply e.g., to changing the observational accuracies (and biases) as well as the number of observations of individual or average elements $x_i(t)$. With these parameters, the change is not large relative to the background, but as a climate analogue, the importance would depend upon the physical significance of a small change (e.g., Wunsch, 2020).

### 1.2.3   The Uncertainties

The structure of the uncertainties depends upon both the model and the detailed nature of the observations. Consider $\mathbf{P}(t)$ for $t = 3523 = m\Delta t$, and one time-step into the future, Fig. 9, just before and after some observations becomes available..





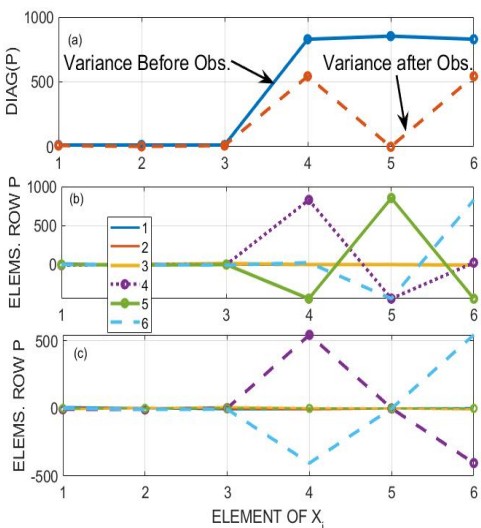

**Figure 9.** Only observation is *velocity* of mass 2. (a) Diagonal element of $\mathbf{P}(t)$ just prior to an observation and after 10 observations of $x_5(t)$ have been obtained. (b,c ) Rows of $\mathbf{P}(t)$, corresponding to the two times in (a), that is before (b) and after (c) observations are used.

Notice that changing variances along the diagonal, and the sometimes strong covariances implied amongst the different elements of $\hat{\mathbf{x}}(t)$ after 10 observations have been used. One of the eigenvalues of $\mathbf{P}(t)$ is almost zero, meaning that $\mathbf{P}(t)$ is singular. In this case, the only observation was relatively accurate—one of the velocity of the second mass. The eigenvector

corresponding to the zero eigenvalue is close to 1 in position 5 (corresponding to the observed $d\xi_2/dt$) and zero elsewhere. The implication is, that because very good observations were made of $d\xi_2/dt$, its uncertainty almost vanishes here, and a weighting of values by $\mathbf{P}(t)^{-1}$, gives it a near infinite weight at that time.

### 1.2.4 A Fixed-Interval Smoother

As already noted, most physical models in use include some form of invariant principles, including quadratic ones related
to energy, linear ones related e.g. to vorticity, or to positions or flows. These principles are violated whenever the model is combined with data. A reasonable inference for science generally is that no system that violates conservation rules for mass or energy etc., can be physically understood in a meaningful way. The need for system descriptions over finite intervals that do satisfy such principles leads to the notion of "smoothers"—in which the state vector over a finite interval satisfies simultaneously a fully-known model, and the data within error bars such that no invariant violation occurs.
The basic notion is to find the corrections, $\mathbf{\Gamma}\tilde{\mathbf{u}}(t)$, the controls, such that the suitably modified prediction model produces a new, the third, state estimate $\tilde{\mathbf{x}}(t, +)$ obeying the model time-evolution while simultaneously, consistent within error bars, of all the data. The plus sign denoting the use of formally future data. In that way, the usual invariants of energy etc., are restored.





$\tilde{\mathbf{x}}(t,+)$ is generally a better estimate than is $\tilde{\mathbf{x}}(t)$ because it "knows" of the occurrence and values of observations *future* to the time $t$ and accounts for them and with potentially important implications for the new estimates of initial conditions. Application

is made below to a slightly more geophysically identifiable example.

The idea of smoothers is again a control theory construct (see Liebelt, 1967, Anderson and Moore., 1979, Brogan, 1991, W06 among many others), and algorithmically a number of different approaches for linear systems have been developed. (Other algorithms exist, including the Lagrange multiplier, iterative least-squares method, used for ECCO by Fukumori et al., 2018, 2019.)

## 2 Barotropic Rossby Waves


Consider now the problem of reconstruction of invariants over the entire time-span of estimation. Realism is still not the goal—rather it is the demonstration of various elements making up estimates in simplified settings in another exercise in geophysical fluid statistics (GFS). A highly idealized oceanographic problem is described.

### 2.1 The Smoothing Problem

A variety of smoothing approaches exists (e.g. Anderson and Moore, 1979; Goodwin and Sin, 1984; Stengel, 1986). Here the "fixed interval" smoother is of most interest. The fundamental idea is straightforward: to find a weighted least-squares fit of the invariant-conserving model (Eq. 2) to the data Eq. (4) at the sampling times. The ECCO Project (e.g., Fukumori et al., 2018, 2019) is a global ocean example of a smoothed solution consistent with the usual invariants, albeit one using a different approach.

### 2.1.1 RTS Smoother


Consider the sequential method of the RTS smoother, in which the assumption is made that the KF has already been used, rigorously, over the finite interval $0 \leq t \leq T_{dur}$, producing the estimates denoted $\tilde{\mathbf{x}}(t,-), \tilde{\mathbf{x}}(t)$ with their corresponding uncertainty covariances $\mathbf{P}(t,-), \mathbf{P}(t)$. At this stage, no further discussion of the data occurs: all information contained in the observations has been exploited by the KF and is encompassed in $\tilde{\mathbf{x}}(t)$ and its uncertainty $\mathbf{P}(t)$. What has *not* been exploited in

an estimate $\tilde{\mathbf{x}}(t), \mathbf{P}(t)$, is the information contained in data that were obtained *afterwards*, $t+m\Delta t > t$, *but that information is present in any later estimates* $\tilde{\mathbf{x}}(t+m\Delta t), \mathbf{P}(t+m\Delta t)$.

The underlying idea is the same as that employed above: if two estimates of state or other variables are available, a better estimate is made by combining them inversely weighted by their uncertainties. The resulting RTS algorithm is more complex appearing than is the KF, because all of the later estimates have a finite correlation with the previous ones, and they cannot

be simply combined without first removing that correlation. (The pure scalar state vector is readily analyzed without any matrix/vector algebra and is written out in Appendix B.). These new estimates become:





$$\tilde{\mathbf{x}}(t,+) = \tilde{\mathbf{x}}(t) + \mathbf{L}(t+\Delta t)\left[\tilde{\mathbf{x}}(t+\Delta t,+) - \tilde{\mathbf{x}}(t+\Delta t,-)\right], \tag{25}$$

$$\mathbf{L}(t+\Delta t) = \mathbf{P}(t)\mathbf{A}(t)^T\mathbf{P}(t+\Delta t,-)^{-1},$$

$$\mathbf{P}(t,+) = \mathbf{P}(t) + \mathbf{L}(t+\Delta t)\left[\mathbf{P}(t+\Delta t,+) - \mathbf{P}(t+\Delta t,-)\right]\mathbf{L}(t+\Delta t)^T, \tag{26}$$

$$\mathbf{Q}(t,+) = \mathbf{Q}(t) + \mathbf{M}(t+\Delta t)\left[\mathbf{P}(t+\Delta t,+) - \mathbf{P}(t+\Delta t,-)\right]\mathbf{M}(t+\Delta t)^T, \tag{27}$$

involving the estimated $\tilde{\mathbf{x}}(t+\Delta t)$, $\mathbf{P}(t+\Delta t,-)$, $\mathbf{P}(t+\Delta t)$ at a formally future time, $t+\Delta t$. As before, he + in the argument is used to label the estimates of these variables as now having employed the formally *future* data. One can examine putative steady-state behavior of the smoothing equations, to the extent it is plausible. Note that an estimate of the control, $\tilde{\mathbf{u}}(t,+)$, is

obtained from the difference between the old forecast estimate, and the value obtained using future data.

### 2.1.2 Example: Rossby Wave Normal Modes

A more recognizably geophysical example than that above is the flat-bottom, linearized $\beta$-plane Rossby wave system, whose governing equation is,

$$\frac{\partial \nabla^2 \psi_1}{\partial t} + \beta \frac{\partial \psi_1}{\partial x} = q(t,x,y), \tag{28}$$

in a square beta-plane basin of horizontal dimension $L$. This problem is taken to be representative of those involving both space and time structures, including boundary conditions. (Spatial variables $x,y$ should not be confused with the state vector or data variables). Eq. (28) and other geophysically important ones are not self-adjoint, and the general discussion of quadratic invariants leads inevitably to adjoint operators (see Morse and Feshbach, 1953 or for bounding problems—Sewell, 1987, Chs. 3,4).

The closed-basin problem was considered by Longuet-Higgins (1964 and later). Pedlosky (1965) and Lacasce (2002) provide helpful discussions of normal modes) and relevant observational data are discussed by Luther (1982), Woodworth et al. (1995), Ponte (1997), Thomson and Fine (2021) and others. The domain is $0 \le x \le L_x$, $0 \le y \le L_y$ with boundary condition $\psi = 0$ on all four boundaries.

Introduce non-dimensional primed variables, $t' = ft, x = Lx', q = q_0 q', \psi_1' = (a^2/f)\psi_1$. Coriolis force $f$, and $\beta$ are evalu-

ated at 30°N. Letting $a$ be the Earth radius, and $\beta = \beta'a/f = 1.7$, the non-dimensional equation becomes,

$$\frac{\partial \nabla^{'2} \psi_1'}{\partial t'} + \beta' \frac{L}{a} \frac{\partial \psi_1'}{\partial x'} = \frac{L^2}{f} q(t',x',y') = q', \tag{29}$$

choosing further $L = a$, and then omitting the primes from here on except for $\beta'$,

$$\frac{\partial \nabla^2 \psi_1}{\partial t} + \beta' \frac{\partial \psi_1}{\partial x} = q(t,x,y). \tag{30}$$



Hairer et al. (2006) describe numerical solution methods that specifically conserve invariants, but these are not discussed here.

This system was used by Gaspar and Wunsch (1989) for a demonstration of sequential estimation using altimetric data. Here a different state vector will be used.

The solution used is the sum over normal modes satisfying the boundary conditions, $\psi_1 = 0$,

$$\psi_1(x,y,t) = \sum_n \sum_m \exp\left(-i\sigma_{nm}t\right) c_{nm} e^{-i\beta' x/\sigma_{nm}} \sin\left(n\pi x\right)\sin\left(m\pi y\right),$$

and obeying the non-dimensional dispersion relation,

$$\sigma_{nm} = -\frac{-\beta'/2}{\sqrt{(n\pi)^2 + (m\pi)^2}}$$

where $c_{nm}$ is a coefficient dependent upon initial conditions and any forcing present; see especially, Pedlosky (1965).

The problem is now made a bit more interesting by addition to $\psi_1$ of a *steady* component, the solution, $\psi_s(x,y)$ from Stommel (1948) whose governing equation in this non-dimensional form is, where $R_a$ is a Rayleigh friction,

$$R_a' \nabla^2 \psi_s + \beta' \frac{\partial \psi_s}{\partial x} = \sin \pi y, \tag{31}$$

$R_a' = R_a/f$, with solution here written in the simple boundary-layer/interior approximation,

$$\psi_s = e^{-x\beta'/R_a'} \sin \pi y + (x-1)\sin \pi y, \tag{32}$$

which leads to a small error in the eastern boundary condition. The $\sin \pi y$ arises from Stommel's assumed time-independent wind-curl.

For the time-dependent components, the state vector is,

$$\mathbf{x}(t) = \text{vec}\left\{c_p(t)\right\},$$

where $p$ is a linear ordering of $n,m$ of total dimension $N \times M = N_{state} - 1$, which is equal to the number of $n$ times the number of $m$, and the state transition equation is,

$$x_j(t+\Delta t) = \exp\left(-i\sigma_j \Delta t\right) x_j(t) + q_j(t), \quad j=1,..,N_{state}-1,$$

with a complex, diagonal state transition matrix, $\mathbf{A}_2 = \text{diag}\left(\exp\left(-i\sigma_p \Delta t\right)\right)$, square of dimension $N_{state} - 1$. A small, nu-

merical dissipation is introduced, multiplying $\mathbf{A}$ by $exp(-bt), b > 0$, to accommodate loss of memory, e.g., as a conventional Rayleigh dissipation. Some special care in computing covariances must be taken when using complex state vectors and transition matrices (Schreier and Scharf, 2010).

The time-independent flow is included as,

$$x_{Nstate}(t+\Delta t) = x_{Nstate}(t), \tag{33}$$

and again,

$$\mathbf{x}(t+\Delta t) = \mathbf{A}\mathbf{x}(t) + \mathbf{B}\mathbf{q}(t), \tag{34}$$





where complex $\mathbf{A}$ is the same as $\mathbf{A}_2$ except with an added zero row and column , and a single non-zero element, $\mathbf{A}(N_{state}, N_{state}) = 1$. Eq. (34) here is taken to exactly describe the putative "truth". $q_{N_{state}}(t) = 0$, because the Stommel solution results from a steady wind.

Consistent with the analysis in Pedlosky (1965), no westward intensification exists in the normal modes, which decay as a whole. Rayleigh friction of the time-dependent modes is permitted to be different from that in the time-independent mean flow—a physically acceptable situation. The value $b = \sigma_{11}/30 = 1.8 \times 10^{-3}$ is used. No particular realism is intended here in the choices of numerical amplitudes, data properties etc. They are chosen only to demonstrate the estimation issues.

Eq. (34) is here taken to be "truth" and to generate the correct fields. As would be necessary in practice, a "prediction" model
is introduced as

$$\mathbf{x}_p(t + \Delta t) = \mathbf{A}\mathbf{x}_p(t) + \mathbf{B}\mathbf{q}_p(t). \tag{35}$$

with the only difference from the truth model in the initial conditions and forcings.

If $\mathbf{q}(t) = 0$ and with no dissipation, then Eq. (28) has several useful invariants: the quadratic invariant of the kinetic energy and of the "energy" in $\psi$—$\mathbf{x}^T(t)\mathbf{x}(t)$ (complex transpose); and the linear invariant of the vorticity or circulation—when
integrated over the entire basin domain. As above, estimates of the quadratic and linear invariants will depend explicitly on initial conditions, forces, distribution and accuracy of the data, and the covariances and bias errors assigned to all of them.

### 2.1.3   System with Observations

A different problem is now posed of determining the transport of the western boundary current (WBC), which is here assumed to be a constant (invariant) in the presence of *both* physical noise—the normal modes—and the random noise of the
observations $\mathbf{y}(\tau_i)$. For determining the transport of the WBC, the presence of both natural noise (the time-dependent modes) and observational noise is analogous to the true physical circulation problem. Non-dimensional normal mode frequencies and periods for $n = 3, 4, 5$, $m = 4, 5, ..., 9$ are shown in Fig. 10. $\Delta t = 29$, $1/b = 553$, $R'_a/\beta' = 0.29$

Initial modal amplitudes are taken to have a slightly "red" property. The field $\psi(t = 167\Delta t)$ is shown in Fig. 11, keeping in mind that apart from the time-mean $\psi$, the structure is the result of a particular set of random forcings.
Noisy observations, $\mathbf{y}(t)$, are taken at the positions in Fig. 11. The prediction model has the correct $\mathbf{A}$, but the magnitudes of the initial conditions are 20% too large, and the forcing field magnitude of $q_p$ is 50% too small (the forcing is complex white noise).

*Aliasing*

In isolation, the observations will time-alias the field, if not taken at minimum intervals of 1/2 the shortest period present
(here $4\Delta t$). A spatial-alias occurs if the separation is less that 1/2 the shortest wavelength present (here $\Delta y = 1/9$). Both these phenomena are present in what follows, but their impact is minimized by the presence of the time-evolution model.



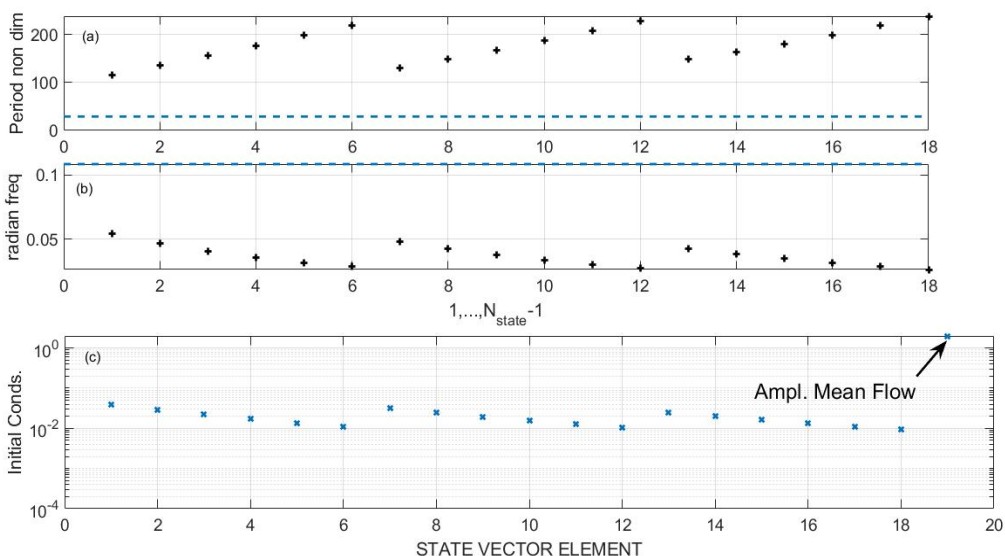

**Figure 10.** (a) Non-dimensional periods, grouped by increasing $n$, and then increasing $m$ for fixed $n$ with $n = 3, 4, 5$, $m = 4, 5, ..., 9$. Dashed line is the computational step, $\Delta t$. (b) Radian frequencies corresponding to the upper panel. (c) Logarithm of the initial conditions for the normal modes.

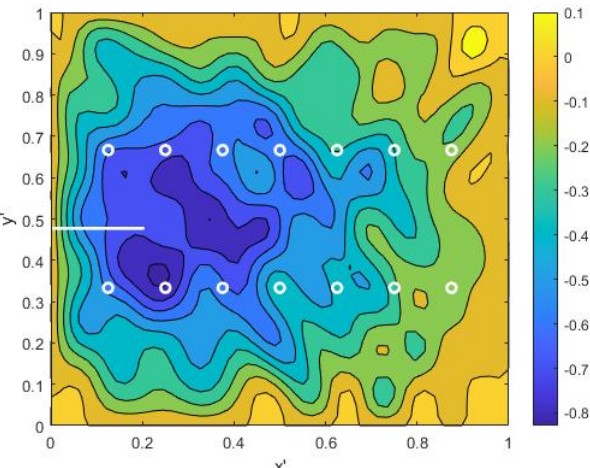

**Figure 11.** Stream function after 11 time steps of including both normal modes and the time–independent Stommel solution. At later times, the mean flow becomes difficult to visually detect in the presence of the growing normal modes under the forcing. White line segment is the distance over which the boundary current transport is defined, slightly shorter than the $e^{-1}$ decay thickness of the boundary current. White circles indicate the assumed 14 available observational positions, fixed in time.





## 2.2 Results: KF+RTS

### 2.2.1 KF Estimates

The RTS algorithm assumes that a proper KF result has been computed and the results stored. $\mathbf{P}(0) = \mathrm{diag}_{Nstate}(1)$, initial
condition uncertainty, and is uniform amongst the elements. Here, as shown in Fig. 12, *a priori* knowledge that higher frequen-
cies have smaller initial values is not being used. $\mathbf{Q}(t) = \mathrm{diag}(0.015)$ except for $x_{Nstate}(t)$ for which $Q_{Nstate,Nstate}(t) =$
$0, \quad \mathbf{R}(t) = \mathrm{diag}_{Nobs}(0.6 \times 10^{-3})$. The system is run with the knowledge that the time-mean wind is truly constant. Obser-
vations are available spatially as in Fig. 11 at intervals, initially at $50\Delta t$ beginning at $t = 166\Delta t$ (Fig. 12) and then more
densely at $25\Delta t$ spacing, a crude mimicking of observations becoming more dense with time. Observations cease prior to
$T_{dur}$, mimicking a pure prediction interval following the observed states.

The energy, $\Phi(t) = \sum_{nm} |\alpha_{nm}(t)|^2$, in the time-dependent components is shown for the true value and the KF estimate
in Fig. 12. It is a surrogate for the total system energy and is a quadratic variable. A slow increase is visible in the true value
and in the prediction value with a levelling off at around time-step 200, again a combination of the dissipation and the white
noise random walk increase. Until the first observation time, the predicted energy is identical to that of the KF, $\Phi_{pred}(t) =$
$\Phi_{kf}(t)$, when the latter takes a jump towards the true value, but remains low. As additional observations accumulate, the
$\Phi_{kf}(t)$ jumps varying amounts depending upon the particulars of the observations and their noise. Over the entire observation
interval the energy remains low—*a systematic* error owing to the sparse observations and null space of $\mathbf{E}$. If the number of
observations is greatly increased (not shown), the systematic error in the estimated energy vanishes. Here the forcing amplitude
overall dominates the effects of the incorrect initial conditions. Uncertainty estimates for energy would once-again come from
summations of correlated $\chi^2$ variables of differing means. In the present case, the most important errors are the systematic ones
visible as the offsets between the curves in Fig. 12.

This system can theoretically be over-determined by letting the number of observations at time $t$ exceed the number of
unknowns—should the null space of $\mathbf{E}(t)$ then vanish. As expected, with 14 covarying observations, and 18 time-varying
unknown $x_i(t)$, $\mathbf{E}(t)$ has a nullspace (rank 12) and thus energy in the true field is missed even if the observations were perfect.
As is conventional in inverse methods, the smaller eigenvalues and their corresponding eigenvectors are most susceptible
to noise biases. The solution nullspace of this particular $\mathbf{E}(t)$ found from the solution eigenvectors of the singular value
decomposition, $\mathbf{U\Lambda V}^T = \mathbf{E}$. Solution resolution matrix at rank $K = 13$, $\mathbf{V}_K \mathbf{V}_K^T$, is shown in Fig. 13 where $\mathbf{V}_K$ contains the
first $K$ columns of $\mathbf{V}$. Thus the observations carry no information about modes (as ordered) 3,6,9,12,15,18. In a real situation,
if control over positioning of the observations was possible, this result could sensibly be modified and/or a strengthening of
the weaker singular values could be achieved. Knowledge of the nullspace structure is very important in the interpretation of
results.

A more general discussion of nullspaces involves that of the weighted $\mathbf{P}(\tau,-)\mathbf{E}^T$ appearing in the Kalman gain. If
$\mathbf{P}(\tau,-)^{1/2}$ is the Cholesky factor of $\mathbf{P}(\tau,-)$ (W06, P. 56), then $\mathbf{EP}(\tau,-)^{1/2}$ is the conventional column-weighting of $\mathbf{E}$
at time $\tau$, and the resolution analysis would be applied to that combination. In the present special case, $\mathbf{A}$ is diagonal, and thus
it does not distribute information about any covariance amongst the elements $x_j(\tau)$ and which would be carried in $\mathbf{P}(\tau,-)$.




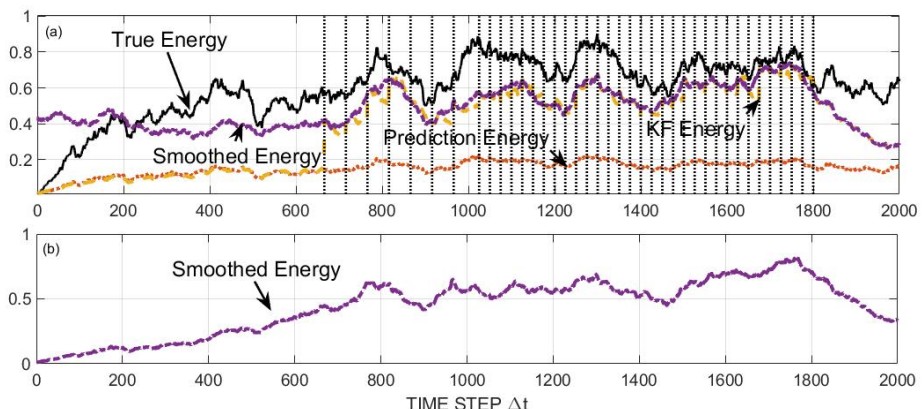

**Figure 12.** (a) The non-dimensional "energy", the sums $\Phi(t) = \sum_{nm} |\alpha_{nm}(t)|^2$, for the true and prediction models of the barotropic system. In the prediction model, the initial conditions are 20% too large, and the forcing is 50% too small (but is otherwise identical to the true value). Time positions of the data, initially at every $50\Delta t$ and then at every 25th $\Delta t$, are shown. Prior to the data onset, the energy is that given by the prediction model. After the data interval, power is also from the prediction model, but starting with the final KF analysis estimate. Jumps in the KF power at observation times are visible, especially at the time of the first observation. The smoothed solution carries too much energy prior to the first observations as the system has no information about the growth of power before that time and the uncertainty assigned to the actual initial conditions is large. (b) The smoothed solution energy when initial conditions are set to be essentially perfect and showing the estimated reduced power towards the origin which does not occur when a finite uncertainty is assigned (as in (a)).

An important observational goal is determination of the north-south transport at each time-step from the velocity or stream function as,

$$WBC(t) = \psi(i=1, j_0, t) - \psi(i_0, j_0, t), \tag{36}$$

at fixed latitude index $j_0$, as the stream-function difference between a longitude pair, $i, i_0$. From the boundary condition, $\psi(i=1, j_0, t) = 0$ identically. In the present context, five different values are relevant: (1) the true constant, invariant, value, (2) the true apparent value including the oscillatory mode noise, (3) the estimated value from the prediction model, (4) the estimate from the KF, (5) the estimate from the smoother. Fig. 14 displays the estimated transport from Eq. (36) for the correct value and from the KF estimate along with the standard error. Values here are dominated by the variability induced by the normal modes. Note that the result can depend sensitively on $i_0, j_0$ and the particular spatial structure of any given normal mode.

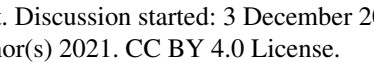





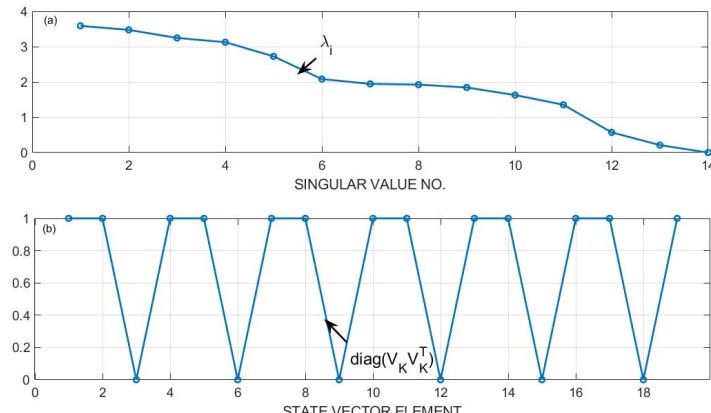

**Figure 13.** (a) First 14 singular vectors of $\mathbf{E}$. Rank is 13 with 14 observational positions. (b) Diagonal elements of the rank 13 solution resolution matrix ($\mathbf{VV}^T$, see W06), showing lack information for several of the modes. A value of 1 means that the mode is fully resolved by the observations. All variables are non-dimensional.

In the KF (Fig. 14), observations move the WBC values closer to the truth, but retain the normal mode noise. Prior to the first observation, the values are indistinguishable from zero. Within the observation interval, the estimates are indistinguishable from the true value, but still have a wide uncertainty with time scales present both from the natural variability and the regular injection times of the data. Transport value uncertainties are derivable from the $\mathbf{P}$ of the state vector.

**2.2.2 RTS Algorithm Results**

Turning now to the RTS smoother, one sees in Fig. 12 that the energy, $\Phi_{smth}(t)$, in the smoothed solution is continuous (up to the usual time-stepping changes), but sometimes exceeds the true energy prior to the appearance of the first observation. The only information available to the smoother prior to the observational interval lies in the initial conditions, which were provided only with a very large uncertainty and the unknown $\mathbf{u}(t)$ in this interval also has a large variance.

The smoother solution in the pre-data interval differs more widely from the true value than does the KF solution. That behavior is a consequence of the comparatively large uncertainty estimate assigned to the initial conditions. If the initial conditions are made near-perfect then (Fig.12), they are reproduced in the smoother solution and the reduced energy in that interval is also the best estimate. An element through time of the control vector and its standard error are shown in Fig. 15. The complex result of the insertion of data is apparent. As with the KF, discussion of any systematic errors has to take place outside

of the formalities leading to the smoothed solution.

Fig. 16 shows the behavior of the estimate of the WBC transport and its uncertainty when the smoothing algorithm is used with two different data densities. A test of the hypothesis that it was indistinguishable from constant would be based upon an analysis using the uncertainty (not shown here).

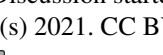


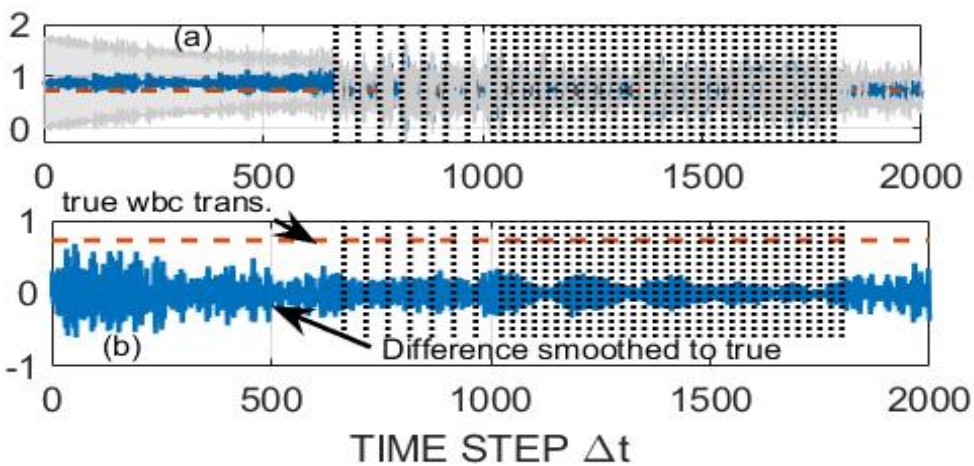

**Figure 14.** (a) Estimated non-dimensional western boundary current transport from the Kalman filter, and its standard error (gray field). (b) Same as (a) except for the smoothed solution and showing the difference between the true value and the estimated one.

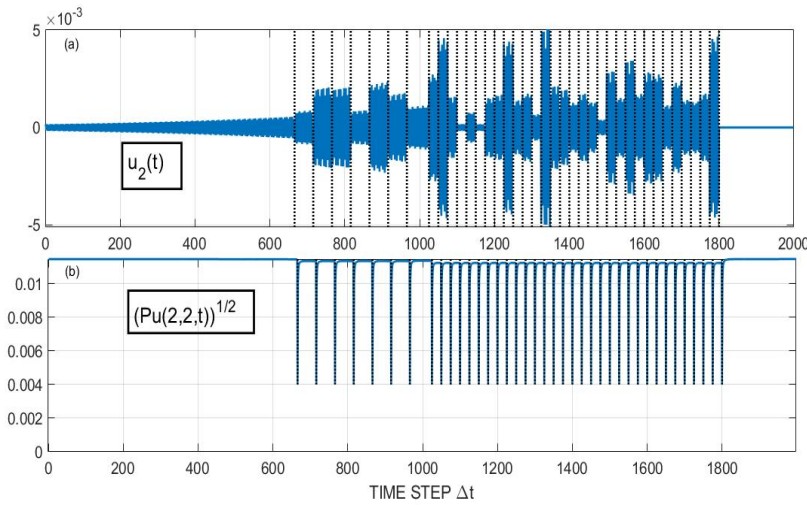

**Figure 15.** (a) One element, $u_2(t)$, of the control vector correction estimate and (b) its standard error through time, showing the drop toward zero at the data time, and the rapid recovery to a higher value.


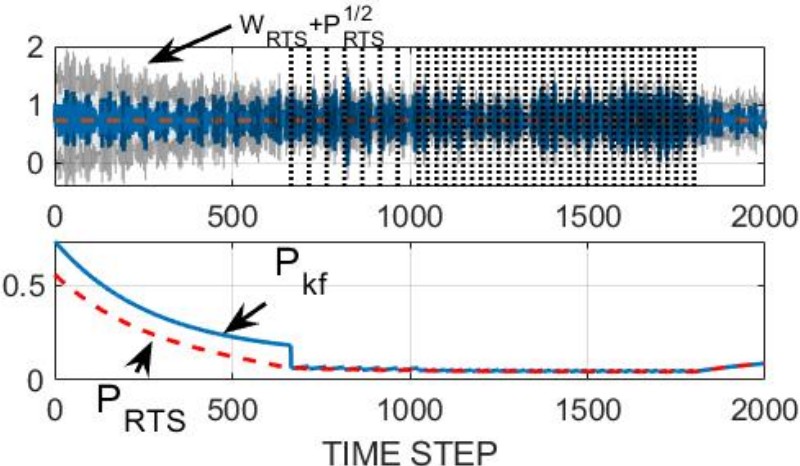

**Figure 16.** (a) Smoothed solution (blue line) estimate of the non-dimensional western boundary current transport and the true mean value (dashed) along with the standard error of the KF estimate (gray lines). Data positions also shown. (b) Uncertainty in the WBC estimate (solid) for the KF and the smoothed values (dashed).

The very large uncertainty prior to the onset of data, even with use of a smoothing algorithm, is a central reason that the
ECCO estimate (e.g., Fukumori, et al., 2018) is confined to the interval following 1992 when the data become far denser than before. Estimates prior to a dense data interval will depend greatly upon the time durations built into the system, which in the present case are limited by the longest normal mode period. The real ocean does include some very long memory (Wunsch and Heimbach, 2008), but the skill will depend directly on the specific physical variables of concern, and which in ECCO include the time-sensitive flow field.

Fig. 17 shows the norm of the operator $\mathbf{L}$ controlling the correction to earlier state estimates, along with the time dependence of one of the diagonal elements. The temporal structure of $\mathbf{L}(t)$ depends directly upon the time constants embedded in $\mathbf{A}$, and the compositions of $\mathbf{P}(t), \mathbf{P}(t+\Delta t, -)$. In turn these latter are determined by any earlier information, including initial conditions, as well as the magnitudes and distributions of later forcing and data accuracies. Generalizations are not easy.

The gain matrix $\mathbf{M}(t)$ for computation of the control vector is displayed in Fig. 18. Here the dependence is directly upon
the a priori known $\mathbf{Q}(t)$, the data distributions, and the determinants of $\mathbf{P}(t+\Delta t, -)$. The limiting cases discussed above for the state vector also provide insights here.

### 2.2.3 Spectra

Computation of the spectral estimates of the various estimates of any state vector element or combination is straightforward and the $z-$transforms of Appendix C provide an analytic approach. What is not so straightforward is the interpretation of




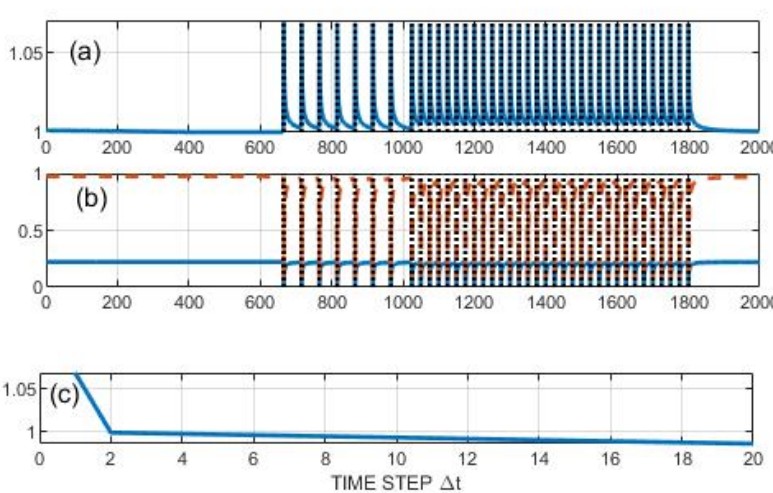

**Figure 17.** (a) Norm of the matrix $\mathbf{L}$ controlling the backwards in time state estimate. (b) Real part (dashed) and imaginary part (solid) component $\mathbf{L}_{22}(t)$ .(c) Norm of $\mathbf{L}$ near the start of the estimation period.

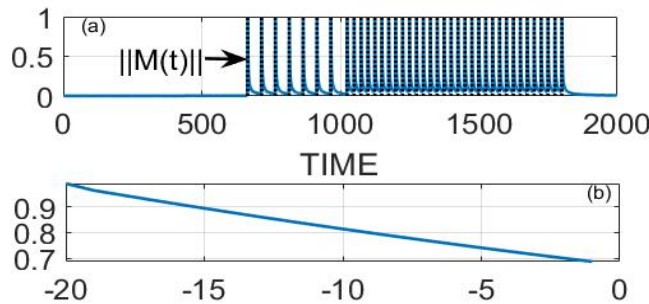

**Figure 18.** (a) Norm of the the gain matrix, $\mathbf{M}(t)$, through time (upper two panels) for the control value.(b) Norm of the products of $\mathbf{M}(t)$ for 20 backwards steps showing strong decrease from the prior observation.



the result in this non-statistically stationary system. Care must be taken to account for the non-stationarity, but results are not further described here.

## 2.3 Discussion

The behavior of dynamical system invariants, be they fundamental ones such as energy or circulation or scalar inventory, or derived ones such as a thermocline depth, in sequential estimation processes depends upon a number of parameters. These
parameters include the time scales embedded in the dynamical system, the temporal distribution of the data relative to the embedded time-scales, the accuracies of initial conditions, boundary conditions, and data, as well as the accuracy of the governing time evolution model. In addition, the invariant estimates can depend directly upon the accuracies of the uncertainties, explicit or implicit, in all of the elements making up the estimation system. Because of their interplay, the only easy generalization is that the user must check the accuracies of all of these elements, including the often difficult appearance of systematic errors
in any of them (e.g., Dee, 2005). When feasible, a strong clue to the presence of systematic errors e.g., in energies, lies in determining the nullspace of the observation matrices coupled with the structure of the state evolution matrices, $\mathbf{A}$. Analogous examples have been computed e.g., for advection-diffusion systems (not shown, but see Marchal and Zhao, 2021) with analogous results concerning e.g., estimates of fixed total tracer inventories.

Physical insights into the system behavior are essential, as is an understanding of the structure of the imputed statistical
relationships. As a considerable literature cited previously has made clear, the inference of trends in properties in the presence of time-evolving observation systems requires particular attention. At a minimum, one should test any such system against the behavior of a known result—e.g., treating a GCM as "truth" and then running the smoothing algorithms to test whether that truth is forthcoming.

## Appendix A: Steady-State and Asymptotics

Time sequence equations starting at $t = 0$ (however defined) undergo a general transient behavior. For simplifying purposes, and following much of the literature, assume that a statistical steady-state has been reached, so that the the innovation equation has become,

$$\tilde{\mathbf{x}}(t) = \mathbf{A}\tilde{\mathbf{x}}(t - \Delta t, -) + \mathbf{B}\mathbf{q}(t - \Delta t) + \mathbf{K}[\mathbf{y}(t) - \mathbf{E}\mathbf{x}(t)], \tag{A3}$$

respectively, with no time-dependence in $\mathbf{A}, \mathbf{B}$, or $\mathbf{K}$. Time-dependence remains in $\tilde{\mathbf{x}}(t)$ but it can be statistically stationary
(labelled "wide" or "weak" depending on the literature). The steady-state error covariance is

$$\mathbf{P}_\infty(-) = \mathbf{A}\mathbf{P}_\infty(-)\mathbf{A}^T + \mathbf{B}\mathbf{Q}\mathbf{B}^T, \tag{A4}$$

$$\mathbf{P}_\infty = \mathbf{P}_\infty(-) - \mathbf{P}_\infty(-)\mathbf{E}^T[\mathbf{E}\mathbf{P}_\infty(-)\mathbf{E}^T + \mathbf{R}]^{-1}\mathbf{E}\mathbf{P}_\infty(\tau, -), \tag{A5}$$





an algebraic Riccati equation, and

$$\mathbf{K}_\infty = \mathbf{P}_\infty\left(-\right)\mathbf{E}^T\left[\mathbf{E}\mathbf{P}_\infty\left(-\right)\mathbf{E}^T+\mathbf{R}\right]^{-1} \tag{A6}$$

is also constant. Pitfalls lie in the accuracies of $\mathbf{P}_\infty$ and in $\mathbf{R}$.

Within a steady-state, the various moments can be computed. So for example, from the innovation state equation, the mean

$$\mathbf{m}_x = \langle\tilde{\mathbf{x}}(t)\rangle = \mathbf{A}\langle\tilde{\mathbf{x}}(t-\Delta t,-)\rangle + \mathbf{B}\langle\mathbf{q}(t-\Delta t)\rangle + \mathbf{K}_\infty\left[\langle\mathbf{y}(t)-\mathbf{E}\mathbf{x}(t)\rangle\right], \tag{A7}$$

or

$$\mathbf{m}_x = (\mathbf{I}-\mathbf{A})^{-1}\left[\mathbf{K}_\infty\langle\mathbf{y}(t)-\mathbf{E}\mathbf{x}(t)\rangle + \mathbf{B}\langle\mathbf{q}(t-\Delta t)\rangle\right] \tag{A8}$$

and thus depends directly upon any bias errors in $\mathbf{y}(t)$ and $\mathbf{E}$, and the accuracy of $\mathbf{K}_\infty$. It will be sensitive directly to the structure and rank of $\mathbf{I}-\mathbf{A}$.

**Predictor-Corrector Methods**

Rigorous Kalman filters are widely used in many applications. In climate systems they are almost never used, despite claims to the contrary, because of the computational cost of Eq. (A1moved). Instead, $\mathbf{K}(\tau)$ is replaced by an ad hoc, often constant, matrix, $\mathbf{K}_{pc}$, and in which Eq. (6) is a predictor-corrector system,

$$\tilde{\mathbf{x}}(\tau) = \tilde{\mathbf{x}}(\tau,-) + \mathbf{K}_{pc}\left[\mathbf{y}(\tau) - \mathbf{E}(\tau)\mathbf{x}(\tau)\right]. \tag{A9}$$

$\mathbf{K}_{pc}$ would be substituted for $\mathbf{K}_\infty$ in the previous equations whether derived from the formal solution Eq. (7) or not. As with the true KF, $\tilde{\mathbf{x}}(\tau)$, once combined with data, using $\mathbf{K}_{pc}$, no longer satisfies the prediction model equations, having undergone a jump in values at time $\tau$. As with the true KF, the predictor-corrector system can be written in innovation form, showing the apparent forcing by data.

Fig. A1 depicts the variation in some elements of the Kalman gain matrix for a set of observations at the places shown. Some elements do tend to become nearly constant at the data times, while others continue to show a structure. Whether choosing $\mathbf{K}_{pc}$ from one particular time is adequate will be very much problem dependent.

**Appendix B: Scalar Systems**

The algebraic statement of the smoother is not easy to penetrate. Consider an even simpler system–that of a scalar obeying a time evolution equation, the "prediction equation" in the above terminology is,

$$\tilde{x}(t+\Delta t) = a\tilde{x}(t) + bq(t) + \Gamma(t)\tilde{u}(t) \tag{C1}$$

where $|a| < 1$ is a constant, and $q(t)$ is a known forcing process. $\Gamma(t)u(t)$ has zero-mean and variance $Q(t)$, perhaps constant in time, and represents any unknown element in $q(t)$. $Q$ is used as $u(t)$ represents the uncertainty in $q$. $b,\Gamma$ are known scale





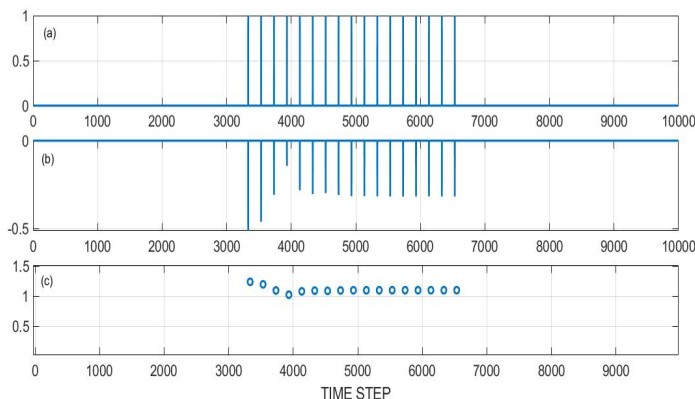

**Figure A1.** Non-dimensional Kalman gain matrix elements reaching steady-state through time and its norm. (a) $\mathbf{K}_{55}(t)$, which is the gain in $x_5(t)$ for an observation of $x_5(t) = \dot{\xi}_2(t)$ .(b) $\mathbf{K}_{65}(t)$—the gain in $x_6(t) = \dot{\xi}_3(t)$ for an observation of $x_5(t)$. (c) $\|\mathbf{K}(t)\|$, the $2$−norm. Only $x_5(t) = \dot{\xi}_2$ was observed. Zero values interlace the observation times.

factors and might as well be taken as $1$, but are useful markers. Let there be observations of $\tilde{x}(\tau)$,

$$y(\tau) = E\tilde{x}(\tau) + \varepsilon(\tau) \tag{C2}$$

where the observation noise, $\varepsilon(t)$, is another zero-mean white noise process of variance, $R$. The system begins at $t = 0$, with an initial condition $\tilde{x}(0)$, with a known uncertainty, $P(0) = (\tilde{x}(0) - x(0))^2$. No null space of $E$ exists

*The Kalman Filter*

Prediction is made using Eq. (C1) with the unknown $u(t)$ set to zero and the estimated initial condition. Then

$$\tilde{x}(t + \Delta t, -) = a\tilde{x}(t) + bq(t). \tag{C3}$$

At the previous time-step, the uncertainty, $P(t) = (\tilde{x}(t) - x(t))^2$, is known, and then the uncertainty of the prediction is,

$$P(t + \Delta t, -) = a^2 P(t) + \Gamma^2 Q(t) \tag{C4}$$

with as before, the minus sign indicating that no data at time $t + \Delta t$ have been used. *If no data are available*, $\tilde{x}(t + \Delta t) = \tilde{x}(t)$, and its uncertainty is $P(t + \Delta t) = P(t + \Delta t, -)$ with Eq. (C4) becoming,

$$P(t + \Delta t) = a^2 P(t) + \Gamma^2 Q(t), \tag{C5}$$

a simple difference equation which can be solved beginning at $t = 0$.



If no data at all are available, taking the $z-$transform, $z = \exp(-2\pi i s \Delta t)$, and denoting the result with a circumflex,


$$\hat{P}(z)/z = a^2 \hat{P}(z) + \Gamma^2 \hat{Q}(z) \tag{C6}$$

and

$$\hat{P}(z) = \frac{z\hat{Q}(z)}{1 - za^2} = \hat{Q}(z)z\left(1 + za^2 + z^2a^4 + ...\right) \tag{C7}$$

If $Q$ is a constant, $\hat{Q}(z) = Q_0\left(1 + z + z^2 + ...\right)$,

$$\hat{P}(z) = Q_0\left(z + z^2 + z3 + ...\right)\left(1 + a^2 z + a^4 z + a^6 z^3 + ...\right) \tag{C8}$$


$$= \frac{zQ_0}{(1 - za^2)(1 - z)}$$

and which will not converge on $|z| = 1$; the pole at $z = 1$ arises from the accumulating influence of the constant $Q$. One might assume a vanishingly small decay constant, $\delta \to 0$, so that $Q(t) = (1 - t\delta)Q_0$ and then,

$$\hat{P}(z) = \frac{z\Gamma^2 Q_0}{(1 - za^2)(1 - (1 - \delta)z)}, \tag{C9}$$

is now interpretable as a Fourier transform on $|z| = 1$.

If data are available at time $\tau$, the weighted average of the two estimates can be written as,

$$\tilde{x}(\tau) = \tilde{x}(\tau, -) + \frac{P(\tau, -)E}{[E^2 P(\tau, -) + R]}(y(\tau) - E\tilde{x}(\tau, -)) \tag{C10}$$

where the difference, $y(\tau) - E\tilde{x}(\tau, -)$ includes both the observational noise, and the discrepancies in the predicted state from the true value. Evidently, any mis-specification of $a, E, Q$ or $P(0)$ will lead to an error in the estimate, and in its uncertainty. With $a < 1$, the influence of initial condition, $\tilde{x}(0)$, will fade with time. In the limit of zero observational noise,


$$\tilde{x}(\tau) = \tilde{x}(\tau, -) + \frac{P(\tau, -)E}{E^2 P(\tau, -)}(y(\tau) - E\tilde{x}(\tau, -)) = y(\tau)/E,$$

as one would expect. In the opposite limit of very large noise, no change is made in $\tilde{x}(\tau, -)$.

The uncertainty following employment of the observation at $t = \tau$ is

$$P(\tau) = P(\tau, -)\left(1 - \frac{P(\tau, -)E^2}{E^2 P(\tau, -) + R}\right) \leq P(\tau, -)$$

that is, the data reduces the uncertainty. Should $R \to 0$, $P(\tau)$ vanishes but would become finite again at the next predicted

time-step. If $R \to \infty$, $P(\tau) = P(\tau, -)$.



Now assuming that the KF has been run out to a duration $0 \leq t \leq T_{dur}$, the Rauch-Tung-Striebel (RTS) algorithm can be used to improve the estimates using observations that were formally future to times $\tau$ in the KF. Let any such new estimate be denoted $\tilde{x}(t,+)$, with a new uncertainty $P(t,+)$. Then for this scalar system,

$$\tilde{x}(t,+) = \tilde{x}(t) + \frac{P(t)\,a}{P(t+\Delta t,-)}\left[\tilde{x}(t+\Delta t,+) - \tilde{x}(t+\Delta t,-)\right] \tag{C11}$$

so that if there were no data *future* to $t+\Delta t$, $\tilde{x}(t+\Delta t,+) = \tilde{x}(t+\Delta t,-)$, and no change is made in the previous value, $\tilde{x}(t)$. If, previously, $\tilde{x}(t)$, were know perfectly, $P(t) = 0$, and again no change is made.

Supposing $\tilde{x}(t+\Delta t,+)$ were perfect e.g., from a perfect observation at that time, $\tilde{x}(t)$ is not simply replaced by the model run backwards, because the change is appropriately partitioned between $\tilde{x}(t,+)$ and $\tilde{u}(t,+)$. The estimated unknown control variable in that interval is,

$$\tilde{u}(t,+) = \frac{Q(t)\,\Gamma(t)}{P(t+\Delta t,-)}\left[\tilde{x}(t+\Delta t,+) - \tilde{x}(t+\Delta t,-)\right] \tag{C12}$$

If $\tilde{x}(t+\Delta t,+)$ is perfect, $\tilde{u}(t)$ is directly proportional to the difference between the predicted $\tilde{x}(t+\Delta t,-)$ and the true value, but not equal to it because some of the change is allotted to $\tilde{x}(t,+)$. Similar constructs can be inferred for the various uncertainties. Should $Q = 0$, $\tilde{u}(t,+) = 0$, and with $P(t+\Delta t,-) = a^2 P(t,)$, then $\tilde{x}(t,+) = \tilde{x}(t) = 1/a\left[\tilde{x}(t+\Delta t,+) - \tilde{x}(\tau+\Delta t,-)\right]$.

**Appendix C:   Green Function Analysis of the Innovation Response**

Define an innovation *matrix,*

$$\mathbf{D}_\delta(t,j) = \mathbf{y}(t) - \mathbf{E}(t)\,\mathbf{x}(t) = \delta_{t,\tau}\boldsymbol{\delta}_{ij} \tag{C1}$$

that is, $\mathbf{D}_\delta$ is a matrix of Kronecker deltas of the difference $D_{ij}(\tau) = \delta_{t,\tau}\boldsymbol{\delta}_{ij} = y_j(\tau) - \sum_r E_{ir}(\tau)x_r(\tau)$. The solutions to the equation are the columns of the Green function matrix,

$$\mathbf{G}(t) = \mathbf{A}\mathbf{G}(t-\Delta t) + \mathbf{K}\mathbf{D}_\delta(t),\ t = m\Delta t. \tag{C2}$$

$\mathbf{K}$, now fixed in time, is sought as an indication of a delta impulse effects of observations on the prediction model at time $\tau$.

Define the scalar complex variable,

$$z = \exp(-i2\pi s\Delta t), -1/2\Delta t \leq s \leq 1/2\Delta t. \tag{C3}$$

Then the discrete Fourier transform of Eq. (C2) (the $z-$transform—a matrix polynomial in $z$) is,

$$\hat{\mathbf{G}}(z) = (\mathbf{I} - z\mathbf{A})^{-1}\mathbf{K}\hat{\mathbf{D}}_\delta(z). \tag{C4}$$

The norm of the variable $(\mathbf{I} - z\mathbf{A})^{-1}$ defines the "resolvent" of $\mathbf{A}$ in the full complex plane (see Trefethen and Embree, 2005), but here, only $|z| = 1$, is of direct interest, that is only on the unit circle. The full complex plane carries information about the behavior of $\mathbf{A}$, including stability.




Here $\hat{\mathbf{D}}(z) = \mathbf{I}z^{\tau}$ and,

$\hat{\mathbf{G}}(z) = (\mathbf{I} - z\mathbf{A})^{-1}\mathbf{K}z^{\tau}$         (C5)

If a suitably defined norm of $\mathbf{A}$ is less than 1,

$\hat{\mathbf{G}}(z) = (\mathbf{I} - z\mathbf{A})^{-1}\mathbf{K}z^{\tau} \approx \left(z^{\tau}\mathbf{I} + z^{\tau+1}\mathbf{A} + z^{\tau+2}\mathbf{A}^2 + z^{\tau+3}\mathbf{A} + ...\right)\mathbf{K}$         (C6)

and the solution matrix in time is the causal vector sequence (no disturbance before $t = \tau$) of columns of

$\mathbf{G}(t) = 0, t < \tau$         (C7)

$= \mathbf{A}^m\mathbf{K}(\tau), t = \tau + m\Delta t$

$m = 0, 1, 2, ...$

$\mathbf{G}$ can be obtained without the $z-$transform, but the frequency content of these results is of interest.

**Green Function of Smoother Innovation**

As with the innovation equation for filtering, Eq. (25) introduces a disturbance into the previous estimate, $\tilde{\mathbf{x}}(t)$, in which the
structure of $\mathbf{L}(t)$ will determine the magnitude and time scales of observational "disturbances" propagated *backwards* in time.
It provides direct insight in the extent to later measurements influence earlier estimates. As one example, suppose that the KF
has been run to time $t = t_f$ so that $\tilde{\mathbf{x}}(t_f, +) = \tilde{\mathbf{x}}(t_f)$, which has the only measurement. Let the innovation, $\tilde{\mathbf{x}}(t_f, +) - \tilde{\mathbf{x}}(t_f, -)$,
be a matrix of $\delta$ functions in separate columns,

$\mathbf{D} = \delta(t - t_f)\mathbf{I}_N$         (C8)

then a backwards-in-time matrix Green function is,

$\mathbf{G}(t) = \mathbf{L}(t)...\mathbf{L}(t_f - \Delta t)\mathbf{L}(t_f)$         (C9)

The various time-scales embedded in $\mathbf{L}$ depend upon those in $\mathbf{A}, \mathbf{P}(t, -), \mathbf{P}(t)$ and with many observations including those
of the observation intervals, and any structure in the observational noise. Similarly, the control modification will be determined
by $\mathbf{P}(t + \Delta t, -)^{-1}$ if $\mathbf{Q}(t)\boldsymbol{\Gamma}(t)^T$ are constant in time.

*Author contributions.* sole author

*Competing interests.* None



*Acknowledgements.* Supported from the NASA/UT Austin/JPL ECCO Projects. Work done at home during the Trump-Covid Apocalypse period.

*Code availability.* Matlab codes used here are available directly from the author.





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
