# Peer review of "Potential Artifacts of Sequential State Estimation Invariants"

_Ocean Science, 2021_

## Referee Comment (RC1)

The author explores potential artifacts in the estimates of physical invariants which are obtained from the quantitative combination of time series data and dynamical models using sequential estimation procedures (Kalman filter and related smoother). Two physical systems are being studied: a system of three coupled oscillators and the wind-driven circulation of a uniform-density fluid in a closed square basin. Emphasis is placed on the determination of "trends" of quantities of oceanographic/climatic relevance, such as mechanical energy and western boundary currents. It is concluded that a robust identification of trends in the studied systems requires detailed understanding of the data, the model, and their respective error covariances, including systematic errors.

The determination of trends in oceanographic time series is of growing interest given the anticipated change in oceanic properties associated with climate change and the increasing availability of time series data from monitoring programs such as RAPID and OSNAP. The study of trends in simplified physical systems such as reported here is essential in order to develop a better understanding of the challenges associated with the determination of trends in more realistic situations involving the analysis of real oceanographic data in the presence of primitive-equation models. The approach applied in the present manuscript - concerned as it is with the estimation of trends in well-understood physical systems - is inspiring. Although the models employed are idealized descriptions of real systems, they permit a discussion of the some of the major issues, such as those associated with the nature of the data, their uncertainties, as well as their spatial and temporal distributions, which are likely to occur in more realistic situations.

The presence of systematic errors in estimates of the mechanical energy of a wind-driven flow, despite the availability of a relatively dense data set, appears to me as one of the most interesting results from this work (Fig. 12). Equally interesting is the analysis of the solution resolution matrix, which permits the identification of dynamical modes that are fully resolved and those that are not resolved, by a given set of observations (Fig. 13). This approach has multiple benefits, e.g., it provides an avenue to test observing strategies, it can add physical considerations to the development of new ones, and it permits a more cautious interpretation of trends in the light of data scarcity (presence of solution nullspace), which is still often a chronic issue in oceanography.

Despite these enthusiastic considerations about the work being presented, I think that a number of comments should be addressed in order for the present manuscript to be suitable for publication.

Major comments are listed below, followed by a list of specific points. I hope this review will help the author to improve his very interesting undertaking.

MAJOR COMMENTS

1) As acknowledged in the manuscript, generalization is difficult, not only because of the idealized character of the models that are employed here but also because of the large number of parameters (e.g., spatial and temporal distributions of the data) that often appear in the quantitative combination of data and models. Nonetheless, I am wondering whether some general results could be obtained by focusing the discussion on the frequency of the observations relatively to the time constants of the dynamical models, for a least a number of data types and/or a number of data spatial distributions. The time constants of the dynamical models, which are here linear, could be obtained from the eigenvalues of the state-transition matrices. It could be instructive to determine the uncertainties of the solution elements (obtained from the KF and/or the RTS) for various recurrence times between observations relatively to the dynamical time constants. To be specific, call the recurrence time between observations $T_{obs}$ and call the eigenvalues of the state-transition matrix $\lambda_1, \lambda_2, \ldots, \lambda_N$. Different filtering and smoothing experiments could be conducted for varying (dimensionless) values of $T_{obs}\,\lambda_1, T_{obs}\,\lambda_2, \ldots, T_{obs}\,\lambda_N$ for different data types (e.g., position or velocity in the coupled oscillator model) and/or different data distributions (e.g., in the wind-driven circulation model). Though a truly exhaustive investigation could be cumbersome, it would be of interest to determine how the variance of the error in the solution elements vary with $T_{obs}\,\lambda_1, T_{obs}\,\lambda_2, \ldots, T_{obs}\,\lambda_N$ for at a least a small number of data types and/or a small number of data spatial distributions.

2) I was intrigued by the suggestion in the manuscript (p. 15, section 1.2.4) that smoother estimates do not violate conservation laws as represented in the model, while Kalman filter estimates do. That KF estimates violate the dynamical equations is obvious from the innovation form of the filter (Eq. 9). On the other hand, that smoothing estimates such as RTS estimates perfectly satisfy the dynamical equations is unclear to me. After all, both the KF and the RTS smoother can be described as weighted least-squares estimates, with the weighting provided by data and model error covariances (see Bryson and Ho, Applied Optimal Estimation, Taylor & Francis, 1975). The chief difference between these 2 methods, as mentioned in the manuscript, is that the state vector

at time $t$ estimated from the KF is not constrained by data posterior to time $t$, whereas the state vector at time $t$ estimated from the RTS is. It is unclear from the smoothing equations (25)-(26) that the RTS equations restore the conservation principles of the dynamical model, for these equations are difficult to penetrate, as nicely phrased in Appendix B. I would like to invite to author to substantiate in the manuscript, preferably via a formal argument, the suggestion that RTS estimates perfectly satisfy conservation principles as represented in the dynamical model.

3) The state-transition matrix $\mathbf{A_c}$ for the system of coupled oscillators (Eq. 14) is not correct. The entry on the $5^{th}$ row, $2^{nd}$ column of $\mathbf{A_c}$ should be -3k, not -2k (see Eq. 11b). The present matrix implies that the restoring force on the $2^{nd}$ mass, or the interaction of the $2^{nd}$ mass with one of the other 2 masses, is not represented. Moreover, the entries including the damping constant $r$ should be *1-r$\Delta$t*, not *(1-r)$\Delta$t*. The corrections should be made in the manuscript and the computer code should be checked. If $\mathbf{A_c}$ is incorrectly coded, the calculations relative to the system of coupled oscillators should be repeated and the associated discussion in the manuscript may need to be modified.

4) I cannot rationalize the early evolution of the KF estimate of mechanical energy in figure 4c (dotted line): since observations are not available during the time interval $0 < t < 5000$, I would expect the KF estimate to be identical to the predicted value during this interval (dashed line in figure 4a). I think this point will need to be clarified in the manuscript.

5) The manuscript is not always easy to follow, and additional details about the assumptions made in the various calculations with the KF and RTS would be welcome. It would be useful to add in the manuscript a table listing the different filtering and smoothing experiments for the system of coupled oscillators, with information about the initial conditions (values and uncertainties), the model parameters (including the time step), the observations (timing and type), and the error covariances assumed for the data and the model. A similar table could be added for the filtering and smoothing experiments for the wind-driven flow in a closed basin. For convenience, each experiment could be labelled (e.g., KF1-O for KF experiment # 1 for the system of coupled oscillators) and these labels could be used in the text for easier reference.

6) The equations for the RTS smoother (Eqs. 25-27) are not correct. The matrix $\mathbf{L}$ in Eqs. (25) and (26) should be the one for time $t$ (not $t + \Delta t$). Likewise, the matrix $\mathbf{M}$ in Eq. (27) should be the one for time $t$, not $t + \Delta t$ (see, e.g., Bryson and Ho, Applied Optimal Estimation, Taylor & Francis, p.

393, 1975). The manuscript should be corrected and the computer code should be checked. If the RTS smoother is incorrectly coded, the calculations should be repeated and the associated discussion in the manuscript may need to be modified.

7) The choice $L = a$ ($a$ = the Earth radius) in the PV equation (29) (line 357) is not consistent with the beta-plane approximation (although the following equation (30) is correct). A choice of $L$ consistent with the beta-plane approximation should be assumed (e.g., Pedlosky J., Geophysical Fluid Dynamics, Springer, 1987).

8) According to my own derivation, both terms on the right-hand side of equation (32) (solution of Stommel model) must be divided by beta'. The corrections should be made in the manuscript and the computer code should be checked. If necessary, the calculations with the corrected code should be repeated and the discussion in the manuscript should be revisited.

9) The multiplication of the state-transition matrix **A** by the factor exp(-b t) (line 380) sounds ad hoc. A more physically-consistent approach would be to add a dissipation term to the PV equation (30) and discretize the resulting term in **A**.

10) The Discussion section (p. 27) is very short and the manuscript lacks a concluding section (e.g., Conclusion or Summary Section). I would suggest to extend, or reorganize, the text, so that an explicit concluding section appears at the end of the manuscript.

SPECIFIC POINTS

Abstract, l. 8: "… more geophysically-relevant problem involving a barotropic …"

L. 42: "… with quantitative models …"

L. 44-47: "Somewhat … (Boers, 2021) suggest … computation, some simple … (Gelaro et al. 2017), clearly …"

L. 55: "… in many other textbooks (notation …"

Footnote on p. 2: "… continuous filter is known …"

L. 65-66: "… true state vector and **u**(t) is a control vector … structure, is introduced …"

L. 78: "… conditions (this sensitivity to initial conditions is typically …"

L. 81: " $= \mathbf{A}(t)\mathbf{P}(t\text{-}\Delta t,\text{-})\mathbf{A}(t)^{\mathrm{T}} + \dots$ "

L. 83-84: "… forcing (elements of $\mathbf{u}(t)$), with $\mathbf{Q}(t) = <\mathbf{u}(t)\mathbf{u}(t)^{\mathrm{T}}>$ (see any … represents the error covariance of u(t). Inaccuracies …"

Footnote on p. 3: "… is restricted to the state vector $\mathbf{x}(t)$ and the control vector $\mathbf{u}(t)$."

L. 89: "… calculate (co)variances …"

L. 93, section *Data*: I suggest replacing $\tau$ by $t$ throughout this section.

Eq. (5a): drop the error term $\mathbf{n}_E(t)$.

Eq. (5b): Put $y$ in bold.

L. 106: "… uncommon, as $\mathbf{x}$ is generally a very large vector and …"

L. 108, section *Combining Data and Models*: I suggest replacing $\tau$ by $t$ throughout this section.

L. 111: "… from equation (xx) above."

L. 125: "… and $\mathbf{E}$, but is omitted here."

Eq. (9) I get

$$\tilde{\boldsymbol{x}}(t + \Delta t) = \boldsymbol{A}(t)\tilde{\boldsymbol{x}}(t, -) + \boldsymbol{B}(t)\boldsymbol{q}(t) + \boldsymbol{K}(t + \Delta t)[\boldsymbol{y}(t + \Delta t) - \boldsymbol{E}(t + \Delta t)\tilde{\boldsymbol{x}}(t + \Delta t, -)]$$

If the above equation is correct, then equation (10) in the manuscript is not.

L. 139-140: Please add parentheses for the references as appropriate.

L. 158: "The 3 x 3 block matrices $\mathbf{K}_c$, $\mathbf{R}_c$ are symmetric and diagonal … distributes inputs $q_c = (q_{c1}, q_{c2}, q_{c3}, 0, 0, 0)^{\mathrm{T}}$." Please also describe $\mathbf{B}_c$.

L. 162: "… are discussed … are omitted …"

L. 165: "Consider … or forcing, and with $\boldsymbol{\xi} = (\xi_1, \xi_2, \xi_3)^{\mathrm{T}}$."

L. 168-169: "Here $\mathcal{E}_c$ is the sum … and is a Hamiltonian."

L. 175: "Discretize equation (11) at constant time intervals $\Delta t$ using …"

Eq. (19): Drop this equation and replace with "where $\boldsymbol{x} = \left(\xi_1, \xi_2, \xi_3, \dot{\xi}_1, \dot{\xi}_2, \dot{\xi}_3\right)^T$." Please also describe $\mathbf{B}$ and $\mathbf{q}$ in Eq. (18).

L. 182-183: "… and necessarily involve …"

Eq. (22), which includes variables for the continuous case, is confusing as it appears in the *Discrete Version* section. Please drop or replace with the discrete analog.

L. 189, "The total energy declines by about 2% in an initial transient …": This statement sounds very anecdotical and a much larger decreases is apparent in figure 2c. Please rephrase or drop.

Eq. (23): Please replace with the discrete analog (and perhaps provide details about the derivation).

Fig. 2: Dashed curve in panel a is not discussed. Panel b is not discussed. Caption: "… $\xi_1(t) - \xi_3(t)$" and typos after "(b)".

L. 204: "… $\mathbf{Bq}(t) = (q_1(t), 0, …, 0)^T$, where $q_1(t) = …$".

L. 211-214: Would this paragraph find a more natural place after the subtitle *Near-Perfect Observations: Two Times and Multiple Times*?

L. 211-212: Please clarify what is meant by "correct"? Write "… deterministic component $q_1(t)$ … stochastic component $\varepsilon(t)$ …".

L. 213-214: Write "… mean. The added … of the total forcing $q_1(t)$". Please rephrase "the latter standard deviation including that of the deterministic contribution."

L. 216: "… of all 6 state variables at two times ..."

Figure 3: Dashed red line in panel b is not discussed.

L. 221: "… Fourier analysis."

Figure 4: Panel b says blue curve is $x_5(t)$ but caption says it is $x_4(t)$. Caption: Please rephrase "Estimated position for velocity in the first mass"

L. 234, "Notice … too low": This result is not clear from figures 4-5.

L. 235: "… availability of observations only of the velocity of one of the masses": I thought $\mathbf{E} = \mathbf{I}_6$ here. Please clarify.

L. 237: "As seen in figure 4" (?)

L. 249: "… with noise similar to …". Here as elsewhere: Please make sure the reader knows which experiment is being discussed (k=?, r=?). Use of labels in a table would help (see major comment).

L. 258-259: "… the observations are very accurate ($\mathbf{R} = xx$) and occur …"

Figure 6, caption: "… and one standard error of the …". Also please use vertical lines, not open circles, to denote the times of data availability for consistency with other figures.

Page 14, first 3 lines: What does it mean for a resolution analysis to be "uniform"? Do you mean the resolution matrix? Next line write "… and of two-mass positions and two-mass velocities …" Similarly, what does it mean for a resolution analysis to be "structured"?

L. 270: "… and $\mathbf{U}_2$ carries the …"

L. 278-279: "… between the estimated and true energy levels over … Compare with …"

L. 282: "… the change of … is not large …". In this sentence, please clarify what the "background" refers to?

Sections 1.2.2 and 1.2.3: It is unclear what particular experiment(s) are being discussed. Use of labels for all experiments, with details about these provided in tables, would resolve the issue.

Figure 9, caption: "… (a) Diagonal elements of …".

L. 288: "… elements of $\tilde{\boldsymbol{x}}(t)$ …" (use a tilde for a consistent notation throughout the manuscript)

L. 308: "… (other algorithms …"

L. 325: "… $t + m\Delta t$ ($m = 1, 2, \dots$), *but that* …"

Eq. 27: Please define $\mathbf{M}(t)$ as well.

L. 337: "… involving the estimates $\tilde{\boldsymbol{x}}(t + \Delta t, -)$ and $\tilde{\boldsymbol{x}}(t + \Delta t)$ and their uncertainties $\boldsymbol{P}(t + \Delta t, -)$ and $\boldsymbol{P}(t + \Delta t)$ …As before, the $+$ …"

L. 342: "… linearized Rossby wave for a uniform-density fluid, whose …". Please specify just after equation (28) what $q(t,x,y)$ is.

L. 351: "… normal modes and relevant …"

L. 354: "…$t' = f_o t$, …, y = Ly', $\psi_1' = \psi_1/(a^2 f_o)$, where $f_o$ is the Coriolis parameter, e.g., at the southern edge of the beta-plane …"

Eq. (29) should have on the right-hand side

$$\left(\frac{L^2}{a^2 f_0^2}\right) q(t', x', y') = q'(t', x', y')$$

where the factor between parenthesis is set to one to ensure that wind forcing is a leading order term in the governing equation.

Page 18: I think the second exponential factor in the definition of $\psi_1'$ should be $\exp\left(-\frac{i\beta' x}{2\sigma_{nm}}\right)$

L. 375, "vec": Please use conventional notation from linear algebra.

L. 376, "… which is equal to the number of n times the number of m": do you mean "n x m"?

L. 379: Drop subscript "2" for matrix **A**.

L. 407: "… We assume $\Delta t = 29, …$ "

L. 415: "… if the separation between observations is greater than …"

Figure 10, caption: "… to the upper panel. Dashed line is … (c) Logarithm …"

L. 420: "… as shown in figure 10 …" (?)

L. 426: $\Phi(t) = \sum_{nm} |c_{nm}(t)|^2$

L. 431: "… jumps by varying …"

L. 440-441: "… the smaller eigenvalues of …" (specify the matrix) ,,, of this particular is found …"

L. 446: "… results and for the development of observing strategies."

Figure 12, caption: $\Phi(t) = \sum_{nm} |c_{nm}(t)|^2$. Replace "KF power" by "energy levels estimated by the KF"

Figure 13, caption: "… showing lack of information …"

L. 470, "The smoother solution … than does the KF solution". This result is not clear from figure 12a.

L. 472: "… are made near-perfect (Fig. 12), then they are …"

Figure 14, caption: "… Kalman filter (blue) … (b) Difference between the true value and the value estimated from the smoothed solution. In both panels, red line shows … and vertical dotted lines show …"

Figure 16, caption: "… (dashed red) along with … of the RTS (?) estimate (gray lines) … (b) Uncertainty in the WBC estimate for the KF (solid) and RTS values (dashed)."

L. 484: "… time-dependent (?) flow field."

L. 485: "… correction to filter state estimates …"

L. 490-491, "The limiting cases discussed above or the state vector also provide insights here": Could you clarify?

Figure 18, caption: "(a) Norm of the gain matrix … through time for the control value."

Figure 18, caption, "Norm of the products of $\mathbf{M}(t)$ for 20 backwards steps showing strong decrease from the prior observation." What are the "products of $\mathbf{M}(t)$"? What specifically is the "prior observation"? (at which time). Please rephrase.

L. 501-502: "… of the dynamical model. In addition, …"

Eq. (A3): last term on the right-hand side should be $K[y(t) - E\tilde{x}(t - \Delta t, -)]$

Eq. (A5): the last matrix on the right-hand side should be $P_\infty(-)$

Eq. (A7): last term on the right-hand side should be $K[y(t) - E\tilde{x}(t, -)]$

L. 535, "(A1moved)"

Eq. (A9): write $t$ instead of $\tau$. Last term on right-hand side should be $K_{pc}[y(t) - E(t)\tilde{x}(t, -)]$

Appendix B: Here and elsewhere: write $t$ instead of $\tau$.

L. 553-554: "… of variance $\mathbf{R}$ … $\mathbf{E}$ exists."

L. 560: "… $\tilde{x}(t + \Delta t) = \tilde{x}(t + \Delta t, -)$ …"

L. 564: "… $z$-transform of …, $z = $ …, where $s$ is …, and denoting …"

L. 581: "… no correction is made to $\tilde{x}(t, -)$."

L. 587, "Rauch-Tung-Striebel": the spelling should also appear in the main text where the acronym RTS first appears.

L. 601: "Define an innovation vector". Please use lower-case for vectors throughout the manuscript. The difference $\mathbf{y}(t)-\mathbf{E}(t)\mathbf{x}(t)$ is simply $\mathbf{n}(t)$. Is this a typo?

L. 603: What is $E_{ir}$?

L. 612: "… only $|z| = 1$ is of …"